# Rapid wavefield forecasting for earthquake early warning via deep sequence to sequence learning

Dongwei Lyu[1,2,3], Rie Nakata[1,2,4], Pu Ren[2], Michael W. Mahoney[1,2,3], Arben Pitarka[5], Nori Nakata[1,2] & N. Benjamin Erichson[1,2] ✉

We propose a deep learning model, WaveCastNet, to forecast high-dimensional wavefields. WaveCastNet integrates a convolutional long expressive memory architecture into a sequence-to-sequence forecasting framework, enabling it to model long-term dependencies and multiscale patterns in both space and time. By sharing weights across spatial and temporal dimensions, WaveCastNet requires significantly fewer parameters than more resource-intensive models such as transformers, resulting in faster inference times. Crucially, WaveCastNet also generalizes better than transformers to rare and critical seismic scenarios, such as high-magnitude earthquakes. Here, we show the ability of the model to predict the intensity and timing of destructive ground motions in real time, using simulated data from the San Francisco Bay Area. Furthermore, we demonstrate its zero-shot capabilities by evaluating WaveCastNet on real earthquake data. Our approach does not require estimating earthquake magnitudes and epicenters, steps that are prone to error in conventional methods, nor does it rely on empirical ground-motion models, which often fail to capture strongly heterogeneous wave propagation effects.

Earthquakes generate complex seismic wavefields as energy is released from the rupture and propagates through the Earth's interior and surface, producing ground motions that can cause significant damage. Real-time prediction of how these ground motions evolve in space and time following the onset of rupture is critical for impact assessment and hazard mitigation, forming the foundation of earthquake early warning systems (EEW). Such forecasts provide immediate information that supports timely and potentially life-saving decision making. However, accurately predicting ground motions remains a challenge. Traditional approaches often rely on estimating earthquake source parameters, such as location, magnitude, and fault geometry, and use empirical ground-motion models to predict shaking intensity[1]. These methods are highly sensitive to errors in early parameter estimates, particularly magnitude, which can result in missed or false alerts by warning systems[2-4]. Moreover, empirical models commonly assume ergodicity, averaging over space and time in ways that neglect important regional and path-dependent variations in wave propagation and site effects[5-9]. Although region-specific or non-ergodic corrections have been proposed[10-13], they are not yet widely adopted.

Recent methods aim to forecast ground-motion intensity measures or waveforms ahead of sensor detection by simulating the evolution of the seismic wavefield[14,15]. These physics-based approaches typically combine numerical simulations (e.g., radiative transfer theory or finite-difference methods) with data assimilation techniques such as optimal interpolation[16]. By directly modeling wave propagation, they avoid reliance on early arrival detection or magnitude estimation and can account for source complexity and path-dependent effects. Despite these advantages, their prediction accuracy has generally been insufficient for operational deployment[1], and the computational cost remains prohibitive for real-time use[15]. These limitations have

[1]International Computer Science Institute, Berkeley, CA, USA. [2]Lawrence Berkeley National Laboratory, Berkeley, CA, USA. [3]Department of Statistics, University of California, Berkeley, CA, USA. [4]Earthquake Research Institute, University of Tokyo, Tokyo, Japan. [5]Lawrence Livermore National Laboratory, Livermore, CA, USA. ✉e-mail: erichson@icsi.berkeley.edu

motivated growing interest in data-driven alternatives that can more efficiently capture the rich spatio-temporal dynamics of seismic wavefields. In particular, deep neural networks have shown promise in modeling ground motions[17–21], and have achieved encouraging results in the context of EEW[22–28]. These models can learn complex spatial and temporal patterns directly from the data, enabling fast and scalable inference suitable for real-time applications.

Building on these recent advances, we formalize the problem of seismic wavefield forecasting within a spatio-temporal sequence prediction framework. The objective is to predict future wavefields based on observed historical data. Formally, we are given a sequence of $J$ elements, $\mathbf{X}_1, \mathbf{X}_2, ..., \mathbf{X}_J$, and aim to predict the subsequent $K$ elements, $\mathbf{X}_{J+1}, \mathbf{X}_{J+2}, ..., \mathbf{X}_{J+K}$. Each element $\mathbf{X}_t \in \mathbb{R}^{C \times H \times W}$ represents a three-dimensional seismic wavefield, encoding particle velocity on an $\mathbb{R}^{H \times W}$ spatial grid across $C$ channels corresponding to the X, Y, and Z directions of wave propagation. Figure 1a–c illustrates our problem setup, including an example snapshot demonstrating viscoelastic wave propagation. Our goal is to forecast future wavefields over time horizons of up to 100 s. However, capturing the complex, multiscale structure inherent in seismic wavefields remains a major challenge.

To address this challenge, we propose a deep learning-based approach for seismic wavefield prediction, as an alternative to the physics-based methods of refs. 14,15, with the aim of reducing

inference time and improving predictive accuracy. Specifically, we develop a wavefield forecasting network, WaveCastNet, that is based on the sequence-to-sequence (Seq2Seq) framework introduced by ref. 29. The core architecture of WaveCastNet consists of an encoder and a decoder: the encoder processes the input sequence of seismic wavefields and summarizes it into a single latent state, while the decoder generates the future sequence conditioned on this state. Figure 1d provides an overview of the WaveCastNet architecture. Both the encoder and decoder are composed of stacked recurrent units designed to handle sequential data.

Modern recurrent cells include unitary recurrent units[30], gated recurrent units (GRUs)[31,32], and long-short-term memory units (LSTM)[33]. However, these architectures rely on fully connected layers, which do not preserve the inherent multiscale spatial information present in 2-D or 3-D data. The convolutional LSTM (ConvLSTM) architecture[34] addresses this limitation by incorporating convolution operations into the update and gating mechanisms of the LSTM, making it particularly effective for modeling spatiotemporal sequences. Although ConvLSTM captures multiscale spatial patterns through convolutional filters, it remains limited in modeling temporal multiscale structures. To overcome this, we design a convolutional long expressive memory (ConvLEM) model, an extension of the LEM architecture[35], that integrates convolutional layers into the LEM

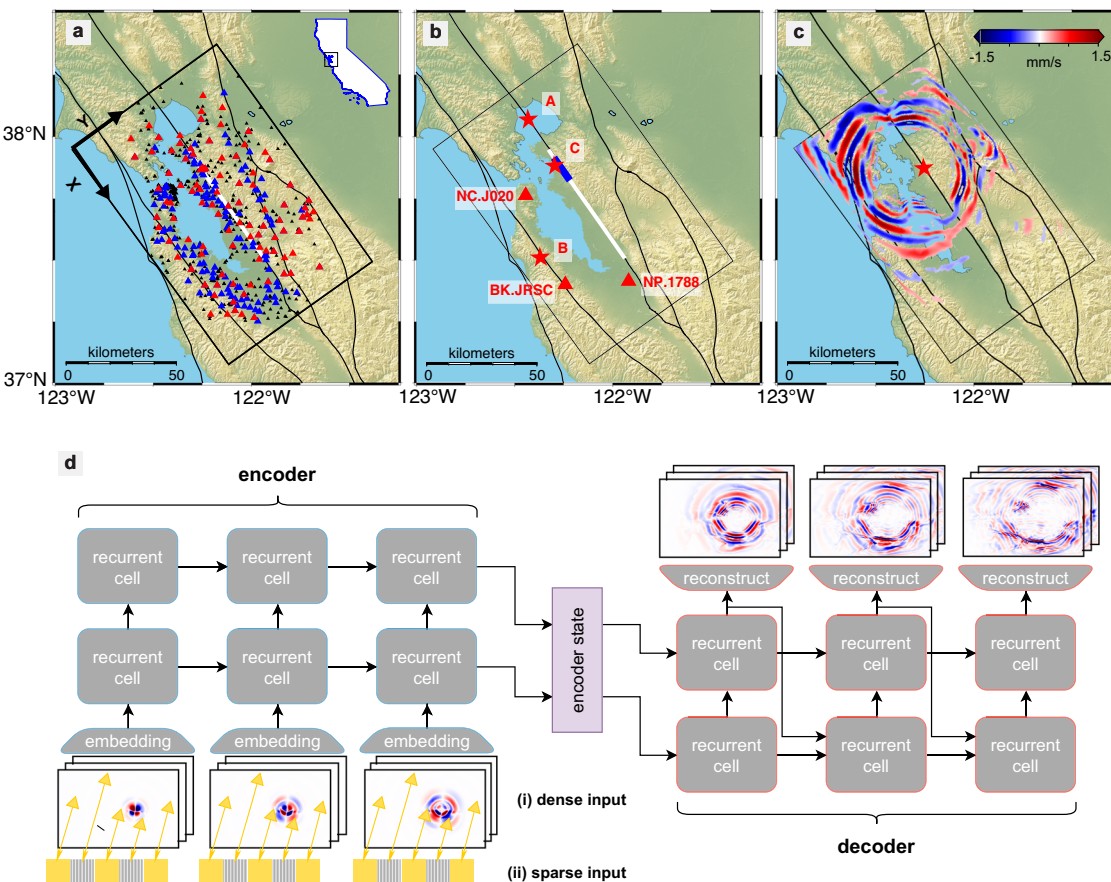

**Fig. 1 | Illustration of the problem setup and our proposed WaveCastNet architecture. a** Simulation domain in the San Francisco Bay Area. The area of interest is indicated by the black rectangular box. Point-source earthquakes are placed along the thick white line. Known faults are shown in black[74]. Black triangles mark the sensor locations used for training under a sparse sensor configuration (see section "Wavefield forecasting network"), red triangles denote the ShakeAlert stations, and blue triangles represent additional sensors used for real-data experiments on the 2018 Berkeley earthquake. Black arrows indicate the X and Y directions of wave propagation. **b** Schematic of a 6.0-magnitude earthquake's

rupture plane (blue line). Red stars indicate epicenters of two hypothetical earthquakes (A, B) and the 2018 Berkeley earthquake (C). Large red triangles highlight the three sensor locations referenced in the discussion. **c** Example snapshot of the horizontal X-component velocity wavefield at $T = 16.67$ s from a point-source simulation. **d** Schematic of the WaveCastNet forecasting framework, consisting of encoder and decoder modules built from stacked recurrent cells. The embedding layer supports both (i) dense wavefield inputs and (ii) sparse sensor inputs, trained using a random masking strategy (gray pixels) to enable high-resolution forecasting.

framework. ConvLEM effectively models the joint spatial and temporal correlations that govern the evolution of the wavefield and serves as the backbone of both the encoder and the decoder in WaveCastNet.

In this work, we show that WaveCastNet can directly model the spatio-temporal evolution of seismic wavefields, enabling real-time prediction of ground motions across a region with high fidelity. Trained on synthetic or observed data, WaveCastNet bypasses the need for explicit arrival detection or source parameter estimation, and produces both full waveform predictions and derived ground-motion intensity measures. These capabilities support detailed wavefield analysis as well as rapid hazard assessment. Once trained, the model operates with low computational overhead, making it suitable for deployment in low-latency EEW systems. We demonstrate robust performance on synthetic benchmarks and present a proof-of-concept using real earthquake data. With further development, including adaptation to observational conditions, validation in various seismic settings, and optimization for subsecond inference, our framework could complement and enhance current ground motion forecasting modules in operational EEW systems.

The main advantages of WaveCastNet are as follows:
- **Predictive Accuracy:** Outperforms Seq2Seq models based on ConvLSTM, gated variants, and transformers.
- **Flexibility:** Capable of handling both dense (fully captured) wavefields and sparse sensor measurements.
- **Generalizability:** Scales to higher-magnitude events and exhibits zero-shot performance on real earthquake data.
- **Robustness to Real-World Conditions:** Maintains performance under background noise and variable data latency.
- **Uncertainty Estimation:** Supports ensemble-based prediction to quantify uncertainty in model outputs.
- **Operational Forecasting:** Forecasts spatially varying waveforms without requiring event detection, source parameter estimation, or empirical ground-motion models.

## Limitations

While our approach demonstrates strong generalization and fast inference on synthetic data, several limitations warrant further investigation. First, although WaveCastNet generalizes to moderate and large-magnitude events, its performance on high-magnitude finite-fault earthquakes remains constrained by the diversity of training data, particularly due to the shift from point sources to extended rupture geometries. This introduces challenges such as amplitude scaling, spatial decay variability, and normalization instability, showing the need to incorporate a broader range of finite-fault simulations during training. Second, our current experiments are restricted to low-frequency waveforms (<0.5 Hz), limiting applicability in scenarios where higher-frequency content is critical, such as near-fault engineering applications. Finally, while we observe promising zero-shot performance on real earthquake data, bridging the domain gap between synthetic and real-world observations remains an open challenge. Addressing this will require expanded datasets, improved data augmentation strategies, and potentially physics-informed regularization to ensure robust performance in operational settings.

## Results

We evaluated our methodology by forecasting particle velocity waveforms near the Hayward Fault, simulating earthquake scenarios in the San Francisco Bay Area as shown in Fig. 1a–c. San Francisco, located ~20 km west of the Hayward Fault, is one of the most densely populated metropolitan areas in the United States. Our prediction domain spans ~120 km along the X direction (parallel to the fault) and 80 km along the Y direction (perpendicular to the fault), delineated by the black rectangle.

We train WaveCastNet using low-magnitude, point-source synthetic waveforms and evaluate its performance across multiple settings. These include both dense, regularly gridded sensor locations and sparse sensor configurations. We further assess the robustness to background noise and test generalization to challenging cases such as out-of-distribution source locations, high-magnitude events, and real earthquake data.

### Point-source small earthquakes

We first evaluate WaveCastNet by predicting ground motions from point-source earthquakes with magnitudes below M4.5. The training dataset is generated using simulated waveforms at frequencies below 0.5 Hz, with a minimum S-wave velocity of 500 m/s. A total of 960 point sources are positioned at 1-km intervals along the white line shown in Fig. 1a. These sources span depths between 2 km and 15 km; the white line denotes a rectangular fault plane extending 60 km horizontally and 13 km vertically. The wavefields are generated using a fourth-order finite-difference viscoelastic model from the open-source SW4 package[36,37], and the subsurface properties are derived from the USGS San Francisco Bay Region 3D seismic velocity model v21.1[38,39]. Each source is modeled as a delta function low-pass filtered at 0.5 Hz, under the assumption that the corner frequencies of small earthquakes exceed this value, resulting in relatively flat frequency spectra within the simulation bandwidth. A uniform double-couple mechanism is applied to all sources. These simulations serve as ground truth for model training and evaluation (see section "Data generation" for details).

WaveCastNet is trained to forecast the next 100 s of the wavefield evolution based on an initial observation window. Rather than generating the entire sequence in a single pass, we adopt an iterative forecasting strategy. Specifically, the model predicts overlapping subsequences of 15.6 s ($J = 60$ time steps at $\Delta t = 0.26$ s), which are recursively fed into subsequent iterations. At each iteration $i$, the model takes as input a subsequence of the form $\{\mathbf{X}_1^i, \ldots, \mathbf{X}_J^i\}$ and outputs the next subsequence $\{\mathbf{X}_1^{i+1}, \ldots, \mathbf{X}_J^{i+1}\}$. Here for each step $k$, $\mathbf{X}_k^i = \mathbf{X}_{(iJ+k)\Delta t}$ and $\mathbf{X}_k^{i+1} = \mathbf{X}_{k+J}^i$. This process is repeated until the entire 100-s forecast horizon is covered. On an NVIDIA A100 GPU, WaveCastNet generates the complete forecast in less than 1 s.

Importantly, the model remains effective even when the initial post-rupture input is shorter than 15.6 s. We can simply pad the shorter sequences with white noise up to the required 60 steps. Accurate forecasts can be achieved with as little as 5.7 s of post-rupture data (see Supplementary Fig. C.1 for how forecasting accuracy evolves over time since rupture onset). In operational settings, this iterative strategy enables continuous real-time forecasting by updating inputs as new observations become available.

In the following, we consider two scenarios: (i) input sequences consisting of densely sampled wavefields; and (ii) input sequences consisting of sparsely sampled wavefields.

**Densely sampled input data.** We begin by evaluating WaveCastNet using densely sampled wavefields as input, where each element of the input and target sequences is a three-dimensional tensor $\mathbf{X}_t$ with dimensions $3 \times 344 \times 224$, corresponding to the three velocity components and the spatial grid in the X and Y directions.

Figure 2a presents a series of ground-truth wavefield snapshots (top row) alongside the corresponding predictions by WaveCastNet (middle row). The model accurately reconstructs the primary characteristics of seismic wave propagation, including the P- and S-wavefronts and scattered coda waves. To further assess performance, we analyze both the intensity and timing of ground motions, focusing on the peak ground velocity (PGV) and its timing ($T_{PGV}$). The spatial distributions of PGV and $T_{PGV}$ are shown in Fig. 2b–d. WaveCastNet accurately reproduces high PGV values and their arrival times, particularly in three key regions: near the earthquake hypocenter at ($X = 40$, $Y = 38$) km, within the Livermore Basin at $X = 60$–$80$ km and $Y = 40$–$60$ km, and in the northeastern portion of the domain at

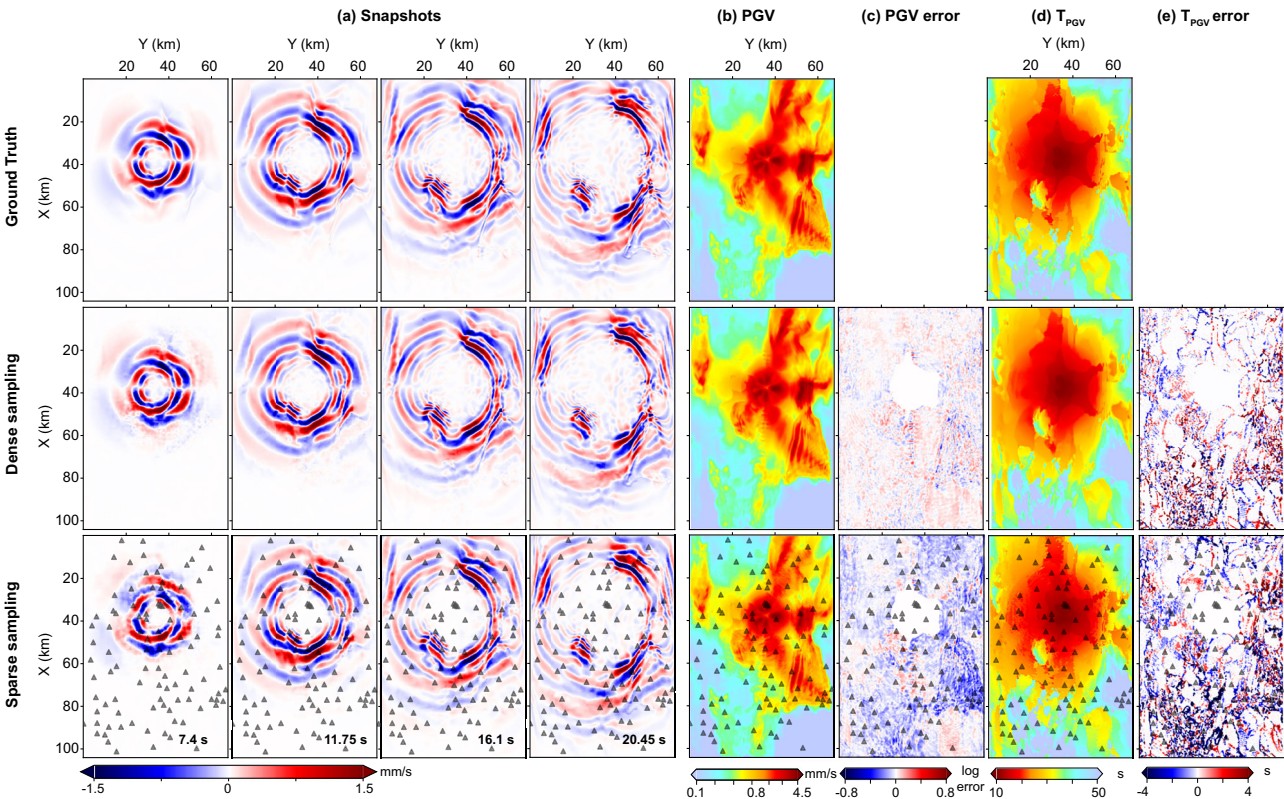

**Fig. 2 | Point-source earthquake prediction. a** Snapshots of Y-component velocity at $T = 7.4, 11.75, 16.1,$ and 20.45 s for (top) ground truth, (middle) predictions from densely and regularly sampled input, and (bottom) predictions from sparsely and irregularly sampled input (triangles denote sensor locations). **b** Map of predicted peak ground velocity (PGV) values and **c** corresponding PGV prediction errors.

**d** Map of predicted PGV arrival time ($T_{PGV}$) values and **e** corresponding $T_{PGV}$ prediction errors. Errors are computed as the difference between predicted and ground-truth values; positive errors indicate overestimation, and negative errors indicate underestimation.

$X = 20\ 40$ km. Deviations in the magnitude of the PGV are minimal, typically within 5% of the ground truth. Errors in $T_{PGV}$ are generally negligible, although localized discrepancies appear in regions where $T_{PGV}$ exhibits discontinuities, likely due to underlying geological complexity.

These results demonstrate that WaveCastNet effectively captures the complex kinematics and dynamics of seismic wave propagation, including amplitude decay from geometrical spreading and intrinsic attenuation, as well as amplification effects due to wave reverberation in geological basins.

**Sparsely sampled input data.** Next we simulate a scenario that more closely reflects real-world conditions, in which seismograph distributions are sparse and irregular, as shown in Fig. 1a. Specifically, we use 101 sensor locations that correspond to stations in the current ShakeAlert network.

To extract sparsely sampled data, we locate the row and column indices $[h, w]$ of each sensor within the wavefield snapshot $\mathbf{X}_t$, forming an input sequence in which each element is a two-dimensional tensor of size $3 \times 101$, corresponding to three velocity components across 101 sparse measurements. The corresponding target sequence consists of densely sampled wavefields, consistent with the previous experimental setup. To accommodate this sparse input representation, WaveCastNet uses a specialized embedding layer that maps sparse measurements to a latent representation, while the rest of the model architecture remains unchanged.

As shown in the bottom row of Fig. 2, WaveCastNet reconstructs wave propagation patterns (PGV, and $T_{PGV}$) across the spatial domain, even when given only sparsely sampled inputs. While the prediction errors are higher than in the densely sampled case, the sparse

measurements still capture the dominant features of seismic wave propagation.

**Uncertainty estimation.** Quantifying uncertainty in ground-motion forecasting is important for risk-informed decision-making. To this end, we adopt an ensemble approach by training 50 instances of WaveCastNet, each initialized with different random seeds and trained on bootstrapped datasets. This setup allows us to compute ensemble means and standard deviations for both time series and their corresponding amplitude spectra in the frequency domain. Figure 3 illustrates the results for the San Francisco and San Jose stations. The ensemble captures the full waveform structure, and the predicted amplitude spectra closely align with the ground-truth data. WaveCastNet also performs reliably in the cases where no seismic energy reaches a station during the initial input window. For example, at the NC.J020 station in San Jose, see Fig. 3b, the model accurately predicts ground motions based solely on contextual information. The ensemble mean values of PGV and their arrival times ($T_{PGV}$) show excellent agreement with the reference data, as shown in Fig. 4a, b, d, e. The standard deviations of the logarithmic PGV and $T_{PGV}$ values are consistently below 1% of their corresponding means, indicating high reliability. Slightly elevated deviations are observed within the Livermore basin, likely due to complex wavefield interactions from multiple incoming fronts, highlighting WaveCastNet 's sensitivity to heterogeneous propagation conditions.

**Generalization**
We assess WaveCastNet 's generalization capabilities across three challenging scenarios: (i) point-source earthquakes located outside the

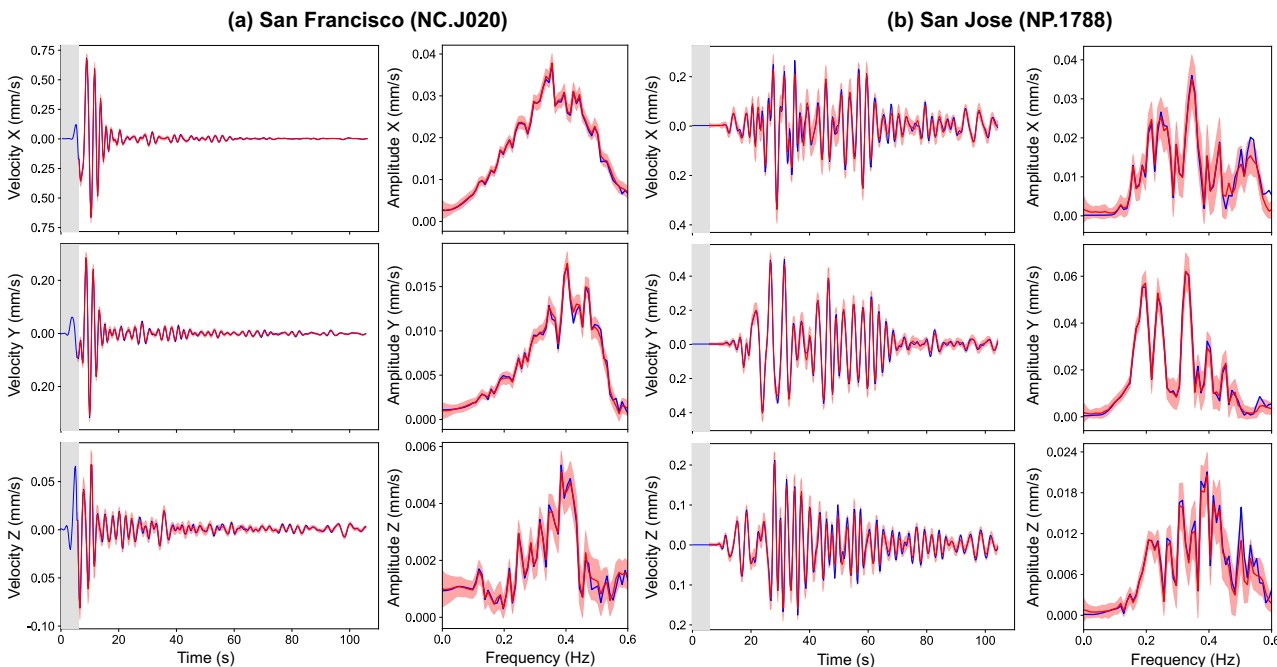

**(a) San Francisco (NC.J020)**   **(b) San Jose (NP.1788)**

**Fig. 3 | Waveform visualization at selected stations.** Waveforms from **a** San Francisco (NC.J020) and **b** San Jose (NP.1788) for a point-source earthquake. The identifiers NC.J020 and NP.1788 refer to seismic sensors. The blue lines indicate the ground truth, the red lines show the mean of the predicted waveforms, and the red shaded areas represent three times the standard deviation bands. The gray shaded areas indicate the input time window of 5.7 s.

training source distribution, (ii) finite-fault, large-magnitude earthquakes, and (iii) real-world earthquake recordings.

**Point sources outside the training distribution.** We first evaluate WaveCastNet on point-source events located beyond the spatial extent of the training dataset. Figure 5 shows PGV and $T_{PGV}$ maps for two such events: one located north of the training region along the Hayward Fault, see point A in Fig. 1b, and another across the Bay along the San Andreas Fault (point B). While WaveCastNet maintains high accuracy for the Hayward Fault event, performance declines for the more distant San Andreas scenario. These results demonstrate that WaveCastNet can generalize wavefield dynamics beyond the training region, but also highlight the expected limitations when forecasting ground motions from events far outside the spatial and structural scope of the training distribution.

**Finite-fault large earthquakes..** Unlike small earthquakes, which can be modeled as point sources, large-magnitude earthquakes require representation as finite rupture planes. Rupture initiates at the hypocenter and propagates along the fault surface, emitting seismic energy from each point over time. This physical process can be approximated by aggregating the responses of many point sources distributed along the fault, each activated at a prescribed rupture time.

Using Green's functions and kinematic rupture models, it is possible to synthesize ground motions for large events by integrating the responses of these point sources[40–43]. Inspired by this approach, we test WaveCastNet 's ability to forecast waveforms from finite-fault earthquakes using only point-source-based training. We employ a suite of kinematic rupture models spanning magnitudes M4.5 to M7 generated using the physics-based method of ref. 42 with updated slip-time functions[43]. These models serve as source terms in SW4 simulations to generate wavefields, a procedure commonly adopted for recorded and scenario earthquakes[43–45]. Each fault rupture is represented by a vertical rectangular plane aligned with the grid of point sources. The rupture dimensions scale with magnitude

following[46], ensuring the appropriate energy release. The source parameters, namely the slip, slip rate, initiation time, and dip, vary spatially and stochastically, resulting in complex ground-motion fields. WaveCastNet does not receive any of these parameters during inference. Furthermore, because the duration of the rupture increases with magnitude[47,48], and the early waveforms remain similar across magnitudes[49], forecasting full wavefields becomes increasingly difficult for larger events.

Initially, we normalize the finite-fault input data using the pixel-wise mean and standard deviation tensors derived from the point-source training set. To improve stability, we further scale inputs using the standard deviation computed from the initial 5.7 s of waveform data. WaveCastNet achieves robust performance for events up to M5.5. However, as shown in Table 1b, performance degrades for M6.0 and larger earthquakes. This degradation correlates with rupture durations, 13.2 s for M6.5 and 26.2 s for M7.0 events, which match or exceed the input window length. As a result, the input does not capture the full duration of the seismic energy release, leading to an underestimation of the amplitude, although the kinematics are still recovered (see Fig. 6a).

To address this limitation, we extended the input time window, an adjustment that is feasible in practical applications. As shown in Figs. 6b–d and 7, this extension improves waveform recovery, particularly low-frequency content. WaveCastNet accurately predicts waveform phases, although it continues to underestimate amplitudes, especially for early arrivals. Despite this, PGV errors remain within 1.5 log units, although timing errors can be significant. In Fig. 6, multiple wavelets exhibit comparable peak amplitudes, complicating their temporal separation, especially in reverberant environments. These findings highlight WaveCastNet 's substantial potential to generalize to finite-fault events, while also identifying areas for further improvement in modeling long-duration, high-magnitude earthquakes.

**Real-world data application.** To assess WaveCastNet 's performance on real-world observations, we evaluate the 2018 *M*4.4

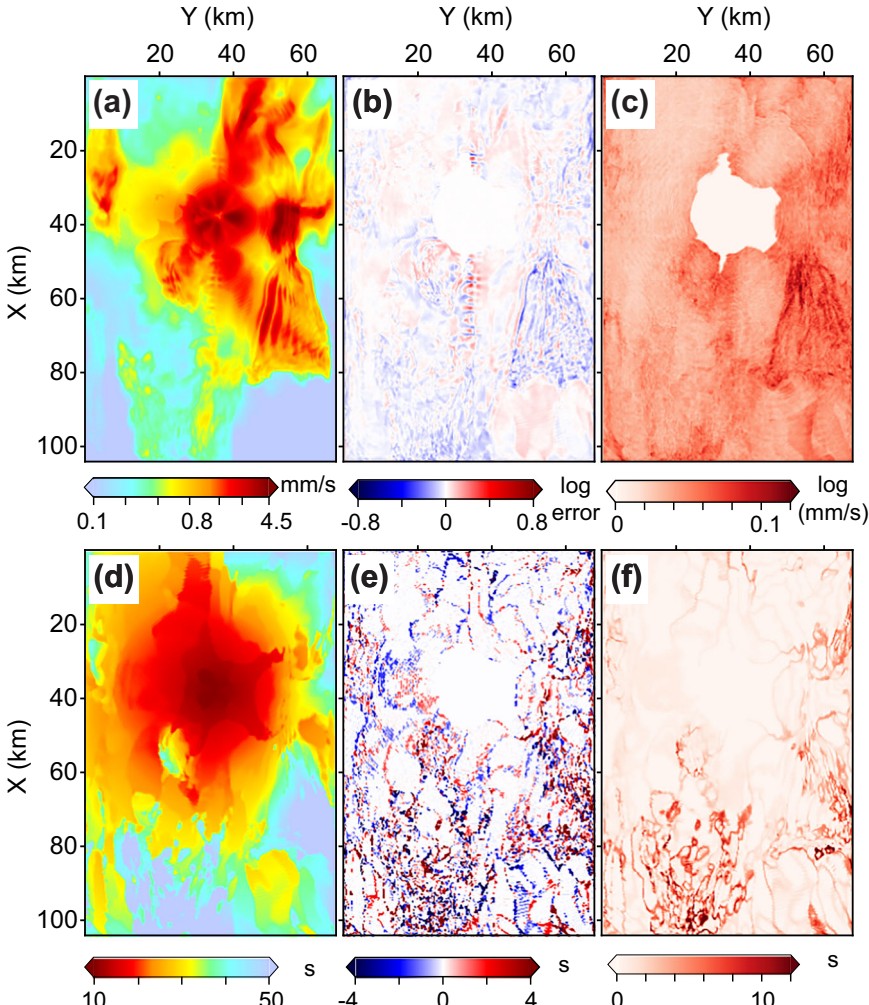

**Fig. 4 | Uncertainty estimates for dense point-source ground-motion prediction. a–c** Mean, error, and standard deviation of the logarithm of peak ground velocity (ln PGV), respectively. **d–f** Corresponding maps for peak ground velocity arrival time ($T_{PGV}$). The error maps **b**, **e** show the difference between predictions and ground truth. The hole in (**f**), centered at $X = 40$, $Y = 38$ km, corresponds to a location where $T_{PGV}$ falls within the input time window and is therefore excluded from evaluation.

Berkeley earthquake, which occurred at a depth of 12.3 km (see section "Data preparation for real world tests"). WaveCastNet was trained exclusively on synthetic data, without any fine-tuning on real waveforms, making this a strict zero-shot generalization scenario. Following the procedure used in our large-magnitude experiments, we apply a 15.6-s input window to predict an equally long output sequence per inference. To improve spatial coverage, we increase the number of input stations from 101 (ShakeAlert) to 178, including additional Berkeley Digital Seismic Network and USGS stations.

As shown in Fig. 8a, the predicted waveforms closely reproduce key characteristics of the observed signals, including shaking duration, prominent S and surface waves, and peak amplitudes. In particular, strong noise in the initial input sequence does not degrade the quality of the forecast, consistent with our earlier robustness tests. While some phase misalignments exist between the predicted and recorded signals, the overall waveform morphology is well captured. These discrepancies likely arise from the mismatch between the true subsurface structure and the velocity model used to generate the synthetic training data[39], as no domain adaptation or fine-tuning was applied.

Figure 8b further shows that WaveCastNet produces spatially coherent PGV and $T_{PGV}$ maps, capturing distance-based attenuation and amplification effects in sedimentary basins. The cross plots in Fig. 8c confirm a strong agreement in the PGV values between the predictions and the observations, while the values $T_{PGV}$ exhibit greater scatter, reflecting the inherent sensitivity of phase-based measurements to structural and timing discrepancies. These results demonstrate WaveCastNet's promise for real-world deployment and highlight the primary challenges associated with zero-shot transfer from synthetic to observational domains.

## Discussion

Our experiments demonstrate that WaveCastNet holds substantial promise for forecasting seismic wavefields derived from both point-source and finite-fault simulations of large-magnitude earthquakes, as well as from real observational data, such as the 2018 $M4.4$ Berkeley event. Across both dense and irregularly sparse sensor configurations, WaveCastNet reliably predicts seismic wave propagation and captures PGV values and their timing with high fidelity. The fit between predicted and ground-truth signals, from initial arrivals to later coda waves, is notable.

We attribute this predictive performance in part to WaveCastNet's ability to internalize the Huygens principle: each spatial point acts as a secondary source of wavefronts. This principle helps explain the model's ability to generalize to out-of-distribution events, including

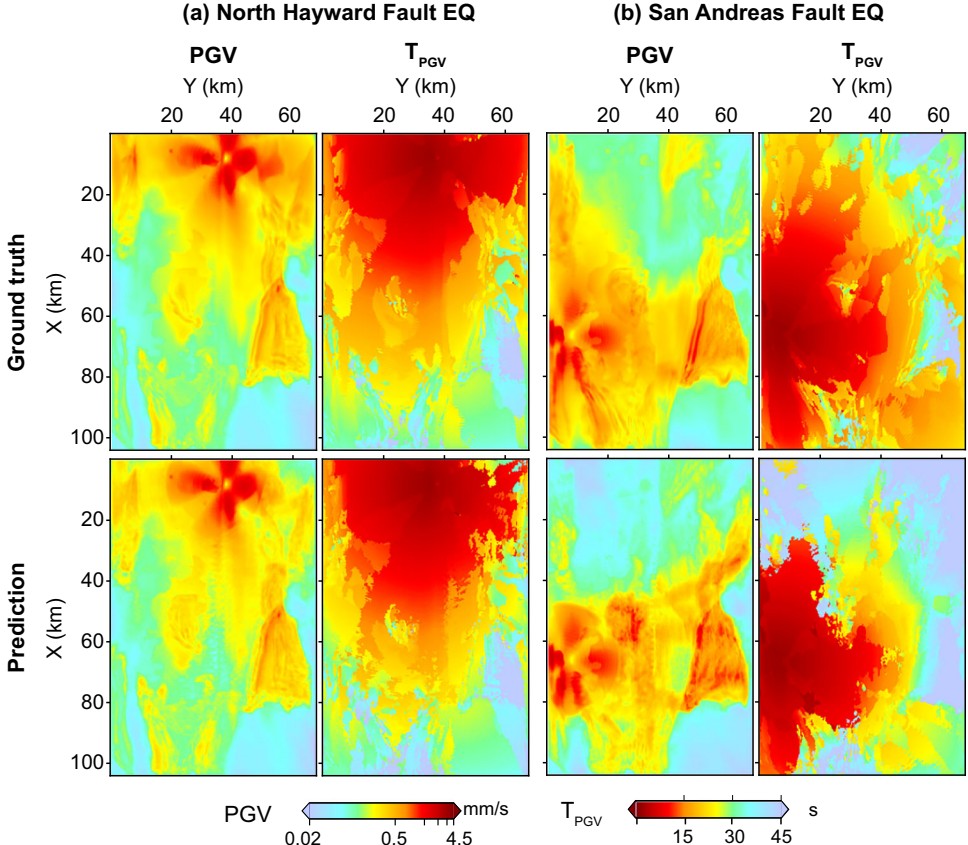

**Fig. 5 | PGV and $T_{PGV}$ predictions for earthquakes located outside the source distribution used during training. a** Event at point A (see Fig. 1b), located north of the training region along the Hayward Fault. **b** Event at point B (see Fig. 1b), located on the San Andreas Fault. For both events: (left) peak ground velocity (PGV) map and (right) PGV arrival time ($T_{PGV}$) map, with (top) ground truth and (bottom) WaveCastNet predictions.

## Table 1 | Model performance summary using 5.7-s input time window

**(a) Performance metrics for the dense and sparse sampling scenarios. Providing the model with more information (i.e., dense inputs) helps to improve performance. ACC (Accuracy) and RFNE (Relative Frobenius Norm Error) are used as evaluation metrics.**

| Input-setting | ACC | RFNE |
|---|---|---|
| Dense and regular sampling | 0.98 | 0.20 |
| Sparse and irregular sampling | 0.93 | 0.36 |

**(b) Fault size and performance metrics of finite-fault earthquake data predictions. $T_{rup}$ indicates the end time of the rupture. See Supplementary Fig. E.1–E.6 for the rupture models.**

| Mw | Fault size (km × km) | $T_{rup}$ (s) | ACC | RFNE |
|---|---|---|---|---|
| 4.5 | 1.8 × 1.8 | 3.5 | 0.95 | 0.35 |
| 5.0 | 3.4 × 3 | 3.7 | 0.95 | 0.37 |
| 5.5 | 8 × 4 | 6.0 | 0.95 | 0.42 |
| 6.0 | 12.5 × 8 | 9.6 | 0.88 | 0.52 |
| 6.5 | 26 × 12 | 13.2 | 0.66 | 0.84 |
| 7.0 | 66 × 15 | 26.6 | 0.53 | 0.86 |

Evaluation on (a) point-source earthquakes with different input sampling strategies; on (b) domain-shifted finite-fault earthquakes across different magnitudes.

point sources located along the San Andreas fault and finite-fault ruptures. In addition to its predictive accuracy, WaveCastNet is robust to background noise, variable input latencies, and diverse sensor configurations (see additional experiments in the Supplementary Section D). In particular, the model generates 100-s forecasts in just

0.56 s on a single NVIDIA A100 GPU, demonstrating its suitability for real-time applications.

Designed to directly predict full waveforms, WaveCastNet effectively captures spatial heterogeneity in ground-motion intensities, an advantage over traditional parameter-based approaches. This opens the door for integration with structural response models, enabling near-instantaneous assessment of infrastructure vulnerability during seismic events.

### Generalization to large-magnitude earthquakes

WaveCastNet generalizes robustly to events up to M5.5, and achieves reasonable accuracy for M6 earthquakes when using a sliding-window inference strategy. This modification, previously used in data assimilation-based early warning systems, accounts for the energy released later in the rupture process and can be implemented without retraining or architectural changes. Importantly, WaveCastNet does not rely on prior knowledge of earthquake magnitude or hypocentral location, suggesting that it can be trained on a relatively limited dataset while still generalizing to a wide range of seismic scenarios.

However, applying a model trained on point-source simulations to large-magnitude events remains a significant challenge. As earthquakes increase in size, their physical representation transitions from point sources to extended rupture planes governed by complex kinematic models. Waveform amplitudes can vary by factors exceeding 80 between M4.5 and M7 events, and spatial decay rates of ground-motion intensity are magnitude-dependent due to differences in fault geometry. These effects complicate data normalization and often lead to underprediction of amplitudes. Our findings show that extending the input window can help improve the forecast accuracy, but this does not fully mitigate these issues. To overcome these limitations, it

**M6.0 Station: NP.1788 Velocity X**

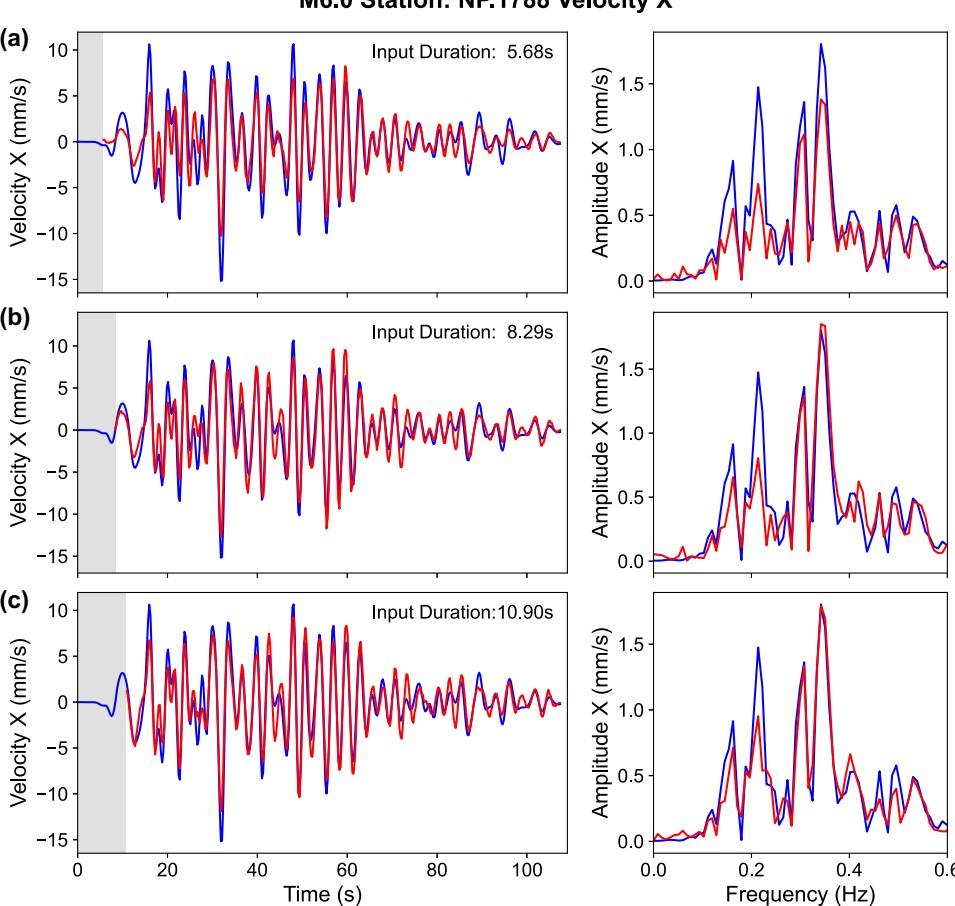

**Fig. 6 | Evolution of ground-motion prediction for a 6.0-magnitude earthquake at station NP.1788 (San Jose), showing the X-component velocity.** Predictions are made using input windows of **a** 5.68 s, **b** 8.29 s, and **c** 10.90 s. The blue lines indicate the ground truth, the red lines indicate the prediction, and the gray shaded regions indicate the duration of the input window used for each inference. Longer input durations lead to improved amplitude recovery in the predicted waveforms.

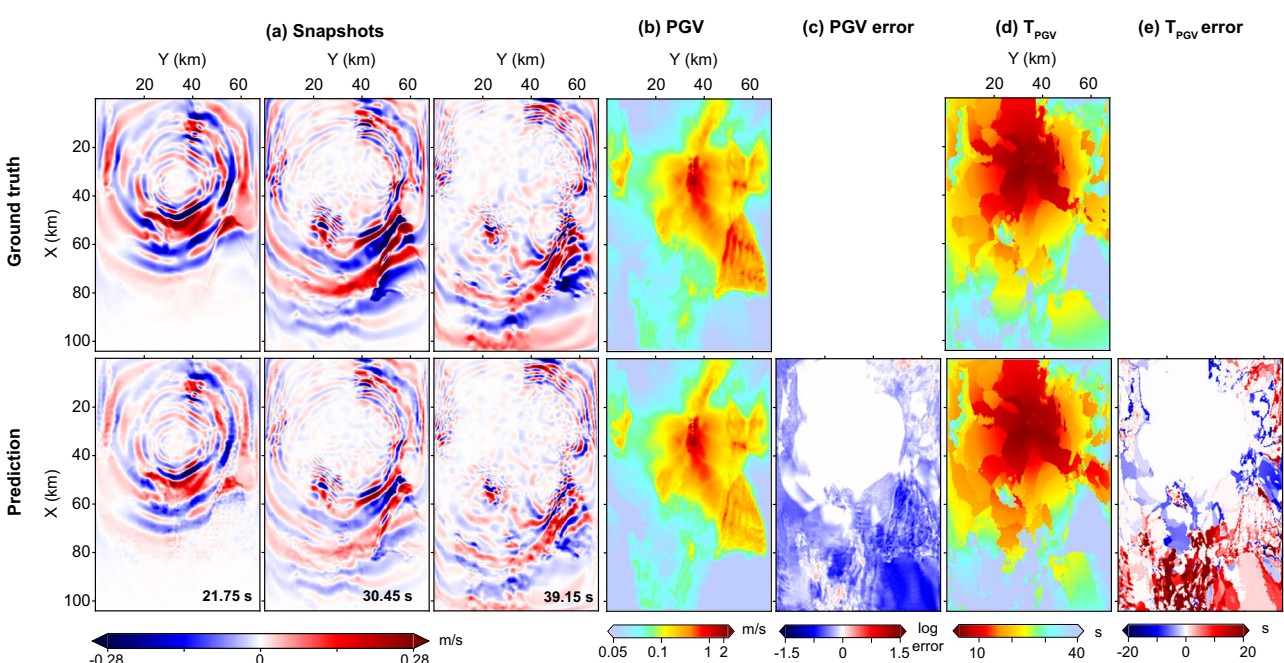

**Fig. 7 | Ground-motion prediction for a 6.0-magnitude earthquake using an 8.2-s input window.** (Top) Ground truth and (bottom) WaveCastNet predictions are shown for each panel. **a** Snapshot of Y-component velocity wavefields at a representative time step. **b** Peak ground velocity (PGV) map. **c** PGV prediction error (predicted minus ground truth). **d** PGV arrival time ($T_{PGV}$) map. **e** $T_{PGV}$ prediction error.

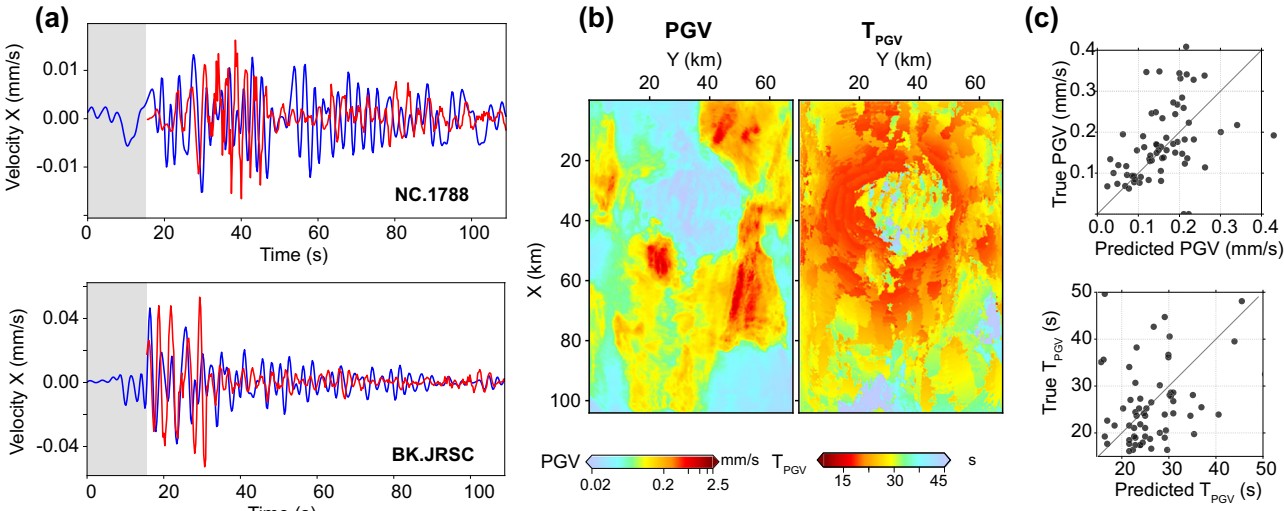

**Fig. 8 | Results for a real-data example in which WaveCastNet is applied to the 2018 Berkeley earthquake (magnitude 4.4, depth 12.3 km). a** X-component velocity waveforms at stations NC.1788 (San Jose) and BK.JRSC (Palo Alto), with WaveCastNet predictions shown in red and observed waveforms in blue. The gray shaded regions indicate the 15.6-s input window used for inference. **b** Spatial maps of predicted peak ground velocity (PGV, left) and its arrival time ($T_{PGV}$, right). **c** Scatter plots comparing predicted and observed values for PGV (top) and $T_{PGV}$ (bottom).

is essential to expand the training dataset to include finite-fault simulations across a broad range of magnitudes. This will enable the model to better learn the amplitude scaling, energy release duration, and rupture complexity inherent in large earthquakes.

## Generalization vs. memorization

Generalization refers to a model's ability to perform well on previously unseen data, while memorization describes the tendency to recall specific patterns or instances from the training set without capturing the underlying relationships[50]. Importantly, memorization is not necessarily detrimental, particularly if the model also generalizes effectively. Recent studies suggest that generalization and memorization often coexist in modern deep learning systems[51], with performance depending on a nuanced balance between the two.

In our case, WaveCastNet demonstrates a successful generalization to out-of-distribution inputs, including large-magnitude finite-fault events, earthquakes originating outside the training domain, and real observational data. Although some degree of memorization may occur, it does not hinder performance; rather, we expect that it coexists with WaveCastNet's ability to learn underlying physical relationships and apply them in various and challenging scenarios.

## Comparative study

To demonstrate the advantages of our proposed approach, we show performance comparisons with baseline models. We evaluated WaveCastNet against Seq2Seq frameworks that use ConvLSTM[34] and ConvGRU[52] as backbones. The results, presented in Fig. 9a, show a lower relative Frobenius norm error (RFNE) for WaveCastNet in the prediction of point-source earthquakes. These experiments use data that are spatially downsampled by a factor of four to ensure model convergence with fewer computational resources (i.e., we use the configuration in section " Point-source small earthquakes", with each $X_t$ reduced to $\mathbb{R}^{C \times \frac{H}{4} \times \frac{W}{4}}$).

Furthermore, we compare WaveCastNet to state-of-the-art transformer architectures designed for spatio-temporal modeling, including the Swin transformer[53,54] and the Time-S-Former[55]. Despite good performance in the task of predicting point source earthquakes, these transformers struggled with generalization in the prediction of higher magnitude earthquakes, as indicated by large relative errors between

magnitudes in Fig. 9b. The comparative study reveals that our WaveCastNet offers beneficial trade-offs: it requires fewer parameters than transformers, enables faster inference times, and introduces a regularization effect through its information bottleneck, aiding generalization.

## Future directions

In the following we discuss future directions that should be studied to extend WaveCastNet. Our experiments mainly used synthetic data at frequencies below 0.5 Hz and demonstrated potential applicability to the real observations. Moving forward, we plan to extend the magnitude and frequency range during the training phase and improve WaveCastNet 's performance on real observations.

- **Extending the Magnitude and Frequency Range**. A key direction for future work is expanding the training dataset to cover a broader range of magnitudes, source complexities, and rupture styles. Our current model is trained on M4-scale point-source simulations, which may not sufficiently capture the physical diversity of larger, finite-fault earthquakes. Incorporating high-magnitude rupture scenarios into the training distribution could improve generalization performance. Although high-frequency synthetic wavefield simulation remains computationally demanding, advances in scalable simulation frameworks and the availability of large open-source simulation databases[44,56,57] may make this direction increasingly feasible.

- **Generative Modeling for Uncertainty-Aware Forecasting**. Future work could also explore extending WaveCastNet with generative frameworks, such as diffusion models, that better reflect the uncertainty in earthquake ground motion forecasting[58,59]. These models may enable the synthesis of full spatio-temporal distributions of wavefield evolution, conditioned on partial observations. Such a probabilistic approach could enhance robustness and offer more physically plausible forecasts across a wide range of magnitudes and source types. However, the effectiveness of these methods for real-time earthquake forecasting remains an open question that needs further investigation.

- **Self-Supervised Pretraining for Foundation Models**. To reduce dependence on large volumes of task-specific labeled data, future research might leverage self-supervised learning (SSL)[60,61] to

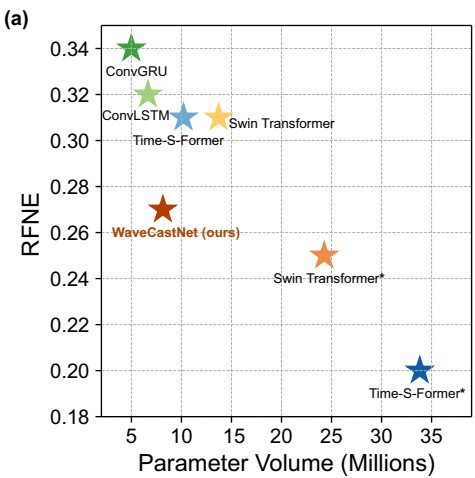

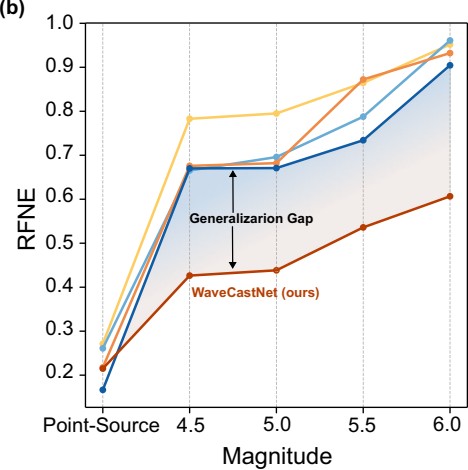

**Fig. 9 | Performance comparison between sequence-to-seqence frameworks with different recurrent cells and state-of-the-art transformers in both in-domain and out-of-domain settings.** The evaluation metric used is RFNE (Relative Frobenius Norm Error). **a** In-domain forecasting performance on point-source earthquakes as a function of model parameter size. **b** Generalization performance as a function of earthquake magnitude. All models are trained only on point-source earthquakes. Colors indicate model type and parameter count (in millions, consistent with (**a**)): yellow for Swin Transformer[53,54] (13.72 million parameters), orange for Swin Transformer* (24.27 million parameters), light blue for Time-S-Former[55] (10.21 million parameters), and dark blue for Time-S-Former* (33.82 million parameters). *indicates larger parameter volume. While larger vision transformers may achieve higher accuracy on in-domain tasks, WaveCastNet generalizes best to domain-shifted settings. Detailed configurations and evaluation metrics for all baselines are provided in Supplementary Table D.1.

pretrain models on unlabeled waveform datasets. Techniques such as masked modeling and temporal contrastive learning[62,63] could help extract generalizable seismic representations. These representations may then be fine-tuned with smaller labeled datasets for downstream tasks like real-world wavefield forecasting, potentially improving data efficiency and enabling deployment in data-sparse regions. The extent to which SSL can improve generalization in this domain is a promising but still largely unexplored area.

- **Fine-Tuning and Domain Adaptation with Real Seismic Data**. Bridging the gap between synthetic training data and real-world deployment remains a critical challenge. Future work should evaluate fine-tuning strategies that adapt pretrained models to real seismic recordings, which exhibit complexities such as sensor noise, site-specific amplification, and regional geological variation. When combined with SSL pretraining or generative modeling, fine-tuning could help calibrate models to the nuances of observational data and improve reliability. However, optimal domain adaptation strategies, especially under limited real data availability, are still an open question that requires further study.

- **Accelerating Inference for Real-Time Forecasting**. To meet the strict latency requirements of early warning systems, future research should also investigate approaches to optimize inference speed without sacrificing accuracy. Promising directions include mixed-precision inference[64], model pruning[65], and quantization[66], which could substantially reduce computational cost. Evaluating the trade-offs between speed, accuracy, and robustness in the context of seismic forecasting remains a necessary step before deployment in a real-world setting.

## Methods

In this section, we describe the methodology underlying WaveCastNet. We begin with an overview of the Seq2Seq framework, which forms the foundation of our forecasting model. We then introduce the ConvLEM architecture that serves as the core building block of WaveCastNet. Finally, we detail the data normalization and preprocessing strategies that we used, as well as the procedures used to generate synthetic datasets and curate the real-world observations used in our experiments.

### Wavefield forecasting network

Similarly to other Seq2Seq models, WaveCastNet is composed of four main components:

- **Embedding layer**. This layer maps input wavefields into a latent space. We employ two types of embedding layers: (i) convolutional layers enhanced with batch normalization and LeakyReLU activation, optimized for embedding densely sampled wavefields into a latent space; and (ii) fully connected layers, followed by convolutional layers, optimized for embedding sparsely sampled wavefields into a latent space. For scenario (ii), we employed a Masked Autoencoder (MAE) strategy, in which 80% of the stations (558 stations recorded motions over the past decade) are randomly masked without replacement following a uniform distribution. The same mask is consistently applied across all time steps within each batch to maintain the continuity of temporal data from active stations. This approach enables a robust embedding layer for sparse sampled data with a significantly reduced number of active stations and mitigates the risk of overfitting that may arise from training on fixed, limited station inputs. This approach can support various sensor configurations.

- **Encoder**. The encoder processes the embedded sequence into a fixed-size encoder state that provides a compressed summary of the input sequence necessary for generating the target sequence.

- **Decoder**. Operating sequentially, the decoder predicts each element of the target sequence one at a time. It uses the previously predicted output combined with the encoder state to forecast the next element.

- **Reconstruction layer**. The reconstruction layer allows us to recover detailed spatial information from predicted latent sequences by using transposed convolutional layers along with pixel-shuffle techniques to reconstruct the high-resolution wavefield.

Both the encoder and decoder use our ConvLEM cell (see section "Convolutional long expressive memory" for details), which is

designed to capture complex multiscale patterns in both spatial and temporal dimensions. Additional technical details of the embedding and reconstruction layers are discussed in the Supplementary Section B.

The Seq2Seq framework seeks to find a target sequence $\mathcal{Y} := \mathcal{X}_{J+1}, \ldots, \mathcal{X}_{J+K}$, from a given input sequence $\mathcal{X} := \mathcal{X}_1, \ldots, \mathcal{X}_J$. The objective is to optimize the conditional probability:

$$\tilde{\mathcal{Y}} = \arg\max_{\mathcal{Y}} p(\mathcal{Y}|\mathcal{X}) \approx \mathcal{D}_{\text{decoder}}(\mathcal{E}_{\text{encoder}}(\mathcal{X})). \quad (1)$$

Although it is challenging to compute the conditional probability directly, an encoder-decoder framework can be used to generate an approximate target sequence[29]. In this process, an encoder, denoted as $\mathcal{E}_{\text{encoder}}$, compresses the embedded input sequence $\mathcal{X}$ into a concise encoder state. Subsequently, a decoder, $\mathcal{D}_{\text{decoder}}$, uses this state to generate the predicted latent sequence $\tilde{\mathcal{Y}}$, which is then mapped to the desired output space by a reconstruction layer. This approach effectively leverages the encoded information to produce a sequence that approximates the target sequence.

WaveCastNet tailors this Seq2Seq framework specifically for the task of forecasting ground motions, treating the prediction challenge as a regression problem. We aim to minimize the sum of all squared differences between the predicted wavefields $\hat{\mathbf{X}}$ and the actual wavefields $\mathbf{X}$:

$$\mathcal{L}_2 = \frac{1}{T}\sum_{t=1}^{T} \| \hat{\mathbf{X}}_t - \mathbf{X}_t \|_F^2, \quad (2)$$

where $\| \cdot \|_F$ denotes the Frobenius norm. Under the assumption that prediction errors follow a normal distribution, minimizing the $\mathcal{L}_2$ loss corresponds to maximizing the likelihood of the data given the model. This approach guides the learning of the model parameters through the minimization of the loss across all actual and predicted sequences. During inference, the model uses these learned parameters to generate target sequences for new input sequences.

To further enhance the model's performance, we adopt the Huber loss during training, defined as follows:

$$\mathcal{L}_{\text{Huber}} = \frac{\sum_{t,c,h,w}\mathcal{L}_\delta\left(\hat{\mathbf{X}}_t[c,h,w], \mathbf{X}_t[c,h,w]\right)}{TCHW}, \quad (3)$$

with the loss function $\mathcal{L}_\delta$ given by:

$$\mathcal{L}_\delta(\hat{x}, x) = \begin{cases} \frac{1}{2}(\hat{x} - x)^2 & \text{for } |\hat{x} - x| \le \delta, \\ \delta \cdot \left(|\hat{x} - x| - \frac{1}{2}\delta\right) & \text{otherwise.} \end{cases} \quad (4)$$

The Huber loss effectively balances the $L1$ and $L2$ norms, which supports a more robust fitting in various earthquake conditions and depths during training. Specifically, we find that the Huber loss improves WaveCastNet's ability to better capture the challenging PGV patterns. Moreover, we observe that using this loss enables our model to better generalize across different earthquake magnitudes and conditions, while also ensuring faster convergence during training.

## Convolutional long expressive memory

We propose ConvLEM to overcome the limitations of traditional recurrent units in modeling complex multiscale structures across spatial and temporal dimensions. These limitations are highlighted when recurrent units are viewed as dynamical systems[67,68], where the evolution over time is governed by a system of input-dependent ordinary differential equations:

$$\frac{d\mathbf{h}}{dt} = \tau \cdot f_\theta(\mathbf{h}(t), \mathbf{x}(t)), \quad (5)$$

where inputs $\mathbf{x}(t) \in \mathbb{R}^d$ and hidden states $\mathbf{h}(t) \in \mathbb{R}^l$ are modeled as continuous functions over time $t \in [0, T]$. However, this model is limited to modeling dynamics at a fixed temporal scale $\tau$. An intuitive approach to address this issue involves integrating a high-dimensional gating function to replace $\tau$, aiming to model dynamics occurring on various time scales[31,32]. However, employing a single gating mechanism often fails to adequately capture the complexities found in more challenging dynamical systems.

In this work, we enhance the modeling of multiscale temporal structures by extending the recently introduced Long Expressive Memory (LEM) unit[35]. This approach is based on the following coupled differential equations:

$$\begin{aligned} \frac{d\mathbf{c}(t)}{dt} &= \mathbf{g}_c \odot \left[ f_{\theta_c}^c(\mathbf{h}(t), \mathbf{x}(t)) - \mathbf{c}(t) \right], \\ \frac{d\mathbf{h}(t)}{dt} &= \mathbf{g}_h \odot \left[ f_{\theta_h}^h(\mathbf{c}(t), \mathbf{x}(t)) - \mathbf{h}(t) \right], \end{aligned} \quad (6)$$

where $\mathbf{h}(t) \in \mathbb{R}^l$ and $\mathbf{c}(t) \in \mathbb{R}^l$ represent the slow- and fast-evolving hidden states, respectively. The gating functions $\mathbf{g}_c$ and $\mathbf{g}_h$, which depend on both input and output states, introduce variability in temporal scales into the dynamics of the model. Here, $\odot$ signifies the Hadamard product, ensuring element-wise multiplication.

We advance the basic LEM unit by incorporating convolutional operations that facilitate modeling of input-to-state and state-to-state transitions, similar to those used in the ConvLSTM model[34]. By representing hidden states and inputs as tensors, we are better able to preserve and model critical multiscale spatial patterns. The ConvLEM is thus formulated as follows:

$$\begin{aligned} \frac{d\mathbf{C}(t)}{dt} &= \mathbf{g}_c \odot \left[ f_{\theta_c}^c(\mathbf{H}(t), \mathbf{X}(t)) - \mathbf{C}(t) \right], \\ \frac{d\mathbf{H}(t)}{dt} &= \mathbf{g}_h \odot \left[ f_{\theta_h}^h(\mathbf{C}(t), \mathbf{X}(t)) - \mathbf{H}(t) \right], \end{aligned} \quad (7)$$

In this equation, $\mathbf{H}(t) \in \mathbb{R}^{r \times p \times q}$ and $\mathbf{C}(t) \in \mathbb{R}^{r \times p \times q}$ denote the slow and fast evolving hidden states, respectively. The input $\mathbf{X}(t) \in \mathbb{R}^{c \times h \times w}$ is a three-dimensional tensor.

To effectively train this model, it is essential to use an appropriate discretization scheme, as it enables the learning of model weights through backpropagation over time. Following the methodology presented in[35], we consider a positive time step $\Delta t$ and use the Implicit-Explicit time step scheme. This approach helps to formulate the discretized version of the ConvLEM unit as follows:

$$\begin{aligned} \Delta\mathbf{t}_n &= \Delta t\, \mathbf{g}_c \\ \overline{\Delta\mathbf{t}_n} &= \Delta t\, \mathbf{g}_h \\ \mathbf{C}_n &= (\mathbb{1} - \Delta\mathbf{t}_n) \odot \mathbf{C}_{n-1} + \Delta\mathbf{t}_n \odot f_{\theta_c}^c \\ \mathbf{H}_n &= \left(\mathbb{1} - \overline{\Delta\mathbf{t}_n}\right) \odot \mathbf{H}_{n-1} + \overline{\Delta\mathbf{t}_n} \odot f_{\theta_h}^h \end{aligned} \quad (8)$$

with update functions

$$\begin{aligned} f_{\theta_c}^c &= tanh(\mathbf{W}_{hc} * \mathbf{H}_{n-1} + \mathbf{W}_{xc} * \mathbf{X}_n), \\ f_{\theta_h}^h &= tanh(\mathbf{W}_{ch} * \mathbf{C}_n + \mathbf{W}_{xh} * \mathbf{X}_n), \end{aligned} \quad (9)$$

and gating functions

$$\begin{aligned} \mathbf{g}_h &= \sigma(\mathbf{W}_{x\bar{t}} * \mathbf{X}_n + \mathbf{W}_{h\bar{t}} * \mathbf{H}_{n-1}), \\ \mathbf{g}_c &= \sigma(\mathbf{W}_{xt} * \mathbf{X}_n + \mathbf{W}_{ht} * \mathbf{H}_{n-1}). \end{aligned} \quad (10)$$

In this notation, $\mathbf{W}_{\cdot\cdot}$ denotes the weight tensors, $\odot$ represents the Hadamard product, and $*$ indicates the convolutional operator, with subscript $n$ marking a discrete time step ranging from 1 to $N$. The matrix of ones, denoted as $\mathbb{1}$, matches the shape of the hidden states. The sigmoid function $\sigma$, used in gating functions, maps activations to a

range between 0 and 1. Note that for brevity, bias vectors are omitted from the update and gating functions.

Based on the model structures outlined above, we further introduce a reset gate $\mathbf{g}_{reset}$ to refine the modeling of the correlation between fast and slow hidden states:

$$\mathbf{g}_{reset} = \sigma\left(\mathbf{W}_{xr} * \mathbf{X}_n + \mathbf{W}_{hr} * \mathbf{H}_{n-1}\right). \tag{11}$$

The reset gate is integrated into the update function for the slow hidden states as follows:

$$f_{\theta_h}^h = tanh\left(\mathbf{g}_{reset} \odot \left(\mathbf{W}_{ch} * \mathbf{C}_n\right) + \mathbf{W}_{xh} * \mathbf{X}_n\right). \tag{12}$$

Intuitively, this additional gate helps to improve the flow of relevant information from the updated fast hidden states to updating the slow hidden states.

Enhancing the gating functions proves beneficial for modeling complex spatio-temporal problems in practice. Using the concept of "peephole connections"[69], we further enhance the gates by injecting information about fast hidden states. We define these gates as follows:

$$\begin{aligned}
\mathbf{g}_h &= \sigma\left(\mathbf{W}_{x\bar{t}} * \mathbf{X}_n + \mathbf{W}_{h\bar{t}} * \mathbf{H}_{n-1} + \mathbf{W}_{c\bar{t}} \odot \mathbf{C}_{n-1}\right), \\
\mathbf{g}_c &= \sigma\left(\mathbf{W}_{xt} * \mathbf{X}_n + \mathbf{W}_{ht} * \mathbf{H}_{n-1} + \mathbf{W}_{ct} \odot \mathbf{C}_n\right), \\
\mathbf{g}_{reset} &= \sigma\left(\mathbf{W}_{xr}\mathbf{X}_n + \mathbf{W}_{hr} * \mathbf{H}_{n-1} + \mathbf{W}_{cr} \odot \mathbf{C}_n\right).
\end{aligned} \tag{13}$$

These modified gates show an improved ability to process longer sequences with more accuracy. Intuitively, by incorporating additional contextual information, these gates are better suited to model complex multiscale dynamics, which in turn improves the model's expressiveness.

## Normalization

Seismic waves exhibit varying residence times as they travel through different geographic locations, leading to significantly greater variance in ground motion in certain regions. Therefore, normalizing is crucial in order to obtain a good forecasting performance. In this work, we use a particle velocity-wise normalization scheme for each snapshot.

Consider all $Q$ sequences in the training set, while each sequence is composed of $T$ snapshots $\{\mathbf{X}_t^q\}$. For each particle velocity $\mathbf{X}[c, h, w]$, we compute the mean and standard deviation values across all snapshots in the training set:

$$\{\mathbf{X}_t^q[c, h, w] \mid q = 0, 1, 2, \ldots Q-1; t = 0, 1, 2, \ldots T-1\}.$$

The resulting mean and standard deviation tensors have the same shape as the snapshot $\mathbf{X}_t$, denoted as $\mathbf{X}_{\text{mean}}$, $\mathbf{X}_{\text{std}}$, respectively. During the data pre-processing stage, for each snapshot $\mathbf{X}_t$, we apply particle velocity-wise normalization as follows:

$$\bar{\mathbf{X}}_t = \frac{\mathbf{X}_t[c, h, w] - \mathbf{X}_{\text{mean}}[c, h, w]}{\mathbf{X}_{\text{std}}[c, h, w]}. \tag{14}$$

Particle velocity-wise normalization also prevents potential spatial information leakage during the normalization process for our sparse sampling scenario.

**Normalization for domain-shifted settings.** The ground-motion of earthquakes with higher magnitudes (e.g., M4.5–M7), once normalized, exhibits a considerably wider range compared to normalized M4 data. Thus, we need to normalize the ground motions again to obtain a reasonable range using the information present in the input window. Given the input window from time step $t_1$ to $t_2$, we perform a channel-wise normalization for each input snapshot $\mathbf{X}_t$ based on the standard deviation values computed for the following set:

$$\begin{aligned}
\{\bar{\mathbf{X}}_t[c, h, w] \mid & t = t_1, t_1+1, \ldots, t_2; h = 0, 1, 2, \ldots H-1; \\
& w = 0, 1, 2, \ldots W-1\}.
\end{aligned}$$

The reasons for not using the particle velocity-wise normalization here are twofold. Firstly, the initial velocity-wise normalization of the particles has already introduced varying standard deviations for different spatial locations. Secondly, since ground motion in the forecast area is observed to be zero within the input window, the velocity-wise standard deviation tensor would consist mostly of zeros, making the normalization process infeasible.

**Metrics.** The performance of WaveCastNet is assessed by analyzing the intensity of ground motions using the PGV values, which are defined as

$$PGV(\mathbf{X}) = \max_t \sqrt{\mathbf{X}_t^2[c_X] + \mathbf{X}_t^2[c_Y]}, \tag{15}$$

where $\mathbf{X}_t^2[c_X]$ and $\mathbf{X}_t^2[c_Y]$ represent the velocity data in the X and Y directions, respectively. Additionally, we examine the corresponding arrival time, $T_{PGV}$, determined by the equation:

$$T_{PGV}(\mathbf{X}) = \arg\max_t \sqrt{\mathbf{X}_t^2[c_X] + \mathbf{X}_t^2[c_Y]}, \tag{16}$$

indicating the moment when the horizontal amplitude of the particle velocity reaches its peak.

Furthermore, to evaluate the accuracy of the predicted wavefield $\hat{\mathbf{X}}$ versus the target ground truth $\mathbf{X}$, we use the accuracy metric (ACC), expressed as:

$$ACC = \frac{\sum_{t,h,w} \hat{\mathbf{X}}_t[c, h, w] \cdot \mathbf{X}_t[c, h, w]}{\sqrt{\left(\sum_{t,h,w} \hat{\mathbf{X}}_t^2[c, h, w]\right) \cdot \left(\sum_{t,h,w} \mathbf{X}_t^2[c, h, w]\right)}}, \tag{17}$$

and the RFNE, defined as:

$$\text{RFNE} = \frac{\sqrt{\sum_{t,h,w} \left(\hat{\mathbf{X}}_t[c, h, w] - \mathbf{X}_t[c, h, w]\right)^2}}{\sqrt{\sum_{t,h,w} \mathbf{X}_t^2[c, h, w]}}. \tag{18}$$

## Data generation

We simulate point-source and finite-fault earthquake ground-motions up to 0.5 Hz within a three-dimensional (3D) volume extending 120 km in the fault parallel (FP) direction (X direction), 80 km in the fault normal (FN) direction (Y direction), and 30 km in depth. These simulations are conducted using the USGS San Francisco Bay region 3D seismic velocity model (SFVM) v21.1[39]. Material properties, including the Vp-Vs relationships, are defined for each geological unit based on laboratory and well-log measurements, which include parameters such as P- and S-wave velocities[39,70,71]. Simulations are initiated with a minimum S-wave velocity of 500 m/s. We generate viscoelastic wave fields using the open source SW4 package, which computes the fourth-order finite difference solution of the viscoelastic wave equations[37]. This software package is well-established, validated through numerous ground-motion simulations[44,45,72].

The surface of the Earth is modeled with a free surface condition, while the outer boundaries use absorbing boundary conditions through a super grid approach spanning 30 grids. We consider a flat surface, and to avoid numerical dispersion, we consider a simulation grid with a mesh size of 150 m³ on the surface, designed to ensure a minimum of six grids per wavelength. To optimize computational resources, the mesh size is doubled at depths of 2.2 km and 6.6 km. The largest grid size used is 600 m³, covering a total of ~9.59 million

grid points. The attenuation and velocity dispersion are modeled using three standard linear solid models, assuming a constant Q over the simulated frequency range. Each simulation runs for 120 s with a time step of 0.0260134 s, resulting in 4613 time steps. The three component particle velocity motions are recorded every 10 steps (i.e., 0.26014 s) on 150 m × 150 m grids and then downsampled to 300 m × 300 m grids for training and testing WaveCastNet.

### Random shifts for latency tests

For simulating the uneven latency test, we introduce random time shifts by sampling from a standard normal distribution $N(0, 1)$. Given a sampled shift value $s$, we compute the adjusted shift as: $\text{sign}(s) \cdot \min(\lceil |s| \rceil, 4)\Delta t$, where $\text{sign}(s)$ preserves the original direction of the shift, $\lceil |s| \rceil$ rounds the absolute value of $s$ to the nearest integer, with the maximum time difference at $4\Delta t$.

### Data preparation for real world tests

Seismic records of the Berkeley 2018 event are downloaded from North California Earthquake Data Center[73]. The downloaded records start 20 s before the reported origin time, and the total recording length is 140 s. After resampling to the 0.01 s time interval, we remove the instrument response and apply a bandpass filter between 0.06 and 0.5 Hz. The data is further downsampled to a time interval of 0.26 s.

## Data availability

The training and evaluation waveform datasets generated in this study are fully accessible via shared cloud drive. Instructions for accessing the data and trained model checkpoints are available on GitHub at https://github.com/dwlyu/WaveCastNet. We provide an efficient data loader for multi-GPU computing in Pytorch. Real waveform data, metadata, or data products are accessed through the Northern California Earthquake Data Center (NCEDC) at doi:10.7932/NCEDC. Quarternary fault traces are obtained from https://www.usgs.gov/programs/earthquake-hazards/faults. San Francisco Bay region 3D seismic velocity model v21.1 used for physic-based simulation is accessed from https://www.sciencebase.gov/catalog/item/61817394d34e9f2789e3c36c.

## Code availability

Research code to reproduce the results of this study is available on GitHub at https://github.com/dwlyu/WaveCastNet. We provide training, evaluation, and visualization scripts in Python, optimized for Pytorch. Physics-based SW4 simulation code is available on GitHub at https://github.com/geodynamics/sw4.

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

## Acknowledgements

N.B.E. acknowledges NSF, under Grant No. 2319621, and the US Department of Energy, under Contract Number DE-AC02-05CH11231 and DE-AC02-05CH11231, for providing partial support of this work. R.N. acknowledges the US Department of Energy Contract No. DE-AC02-05CH11231 and DESC0016520 for providing partial support. AP's work was performed at the Lawrence Livermore National Laboratory under contract DE-AC52-07NA27344. M.W.M. acknowledges the NSF and the DOE, under the LBNL's LDRD program. This research used computational cluster resources, including LLNL's HPC Systems (funded by the LLNL Computing Grand Challenge), LBNL's Lawrencium and NERSC (under Contract No. DE-AC02-05CH11231).

## Author contributions

D.L., R.N. and N.B.E. conceived the study and designed the research plan. D.L. performed the numerical simulations and data analysis. R.N. and N.N. contributed to the seismological interpretation and methodological guidance. P.R. supported model development and implementation. M.W.M. provided input on algorithmic design and theoretical underpinnings. R.N. performed the physics-based seismic modeling, and A.P. provided earthquake rupture models. N.B.E. supervised the computational aspects of the study and coordinated project integration. All authors contributed to the interpretation of results and the writing of the manuscript. R.N. and N.B.E. co-led the project and share senior authorship.

## Competing interests

The authors declare no competing interests.
