## [Transparent Peer Review file · Nature Communications]

Rapid Wavefield Forecasting for Earthquake Early Warning via Deep Sequence to Sequence Learning

Corresponding Author: Dr N. Benjamin Erichson

Version 0:

Reviewer comments:

Reviewer #1

(Remarks to the Author)

Dongwei Lyu et al. have developed a deep learning framework for earthquake early warning. First, they create a database of simulated ground motion fields using small point source earthquakes. Deep learning models are then trained to forecast ground motion 100 seconds in the future, using a deep learning architecture that they specifically designed for the task, and outperforms other more generic architectures.

The results they show are convincing, with the trained deep learning models able to very accurately predict ground motion, peak ground velocity and time of arrival of the strongest shaking.

However, I have three main concerns that I believe need to be addressed in order for the paper to be considered for publication in Nature Communications:

1) The paper does not elaborate at all about how the deep learning models are assessed in terms of training and testing. In particular, if testing and training rupture scenarios are assigned randomly, while all the sources are distributed along a dense line, all the model has to do is to interpolate between two nearby cases, and the model can simply 'memorize' the data. This concern could be alleviated by having test data coming from clearly distinct scenarios, for example the southern half of the fault used for training and the northern half used for testing.

2) One of the most critical tasks for EEW is to correctly alert for larger earthquakes, that are rarer and for which EEW system always perform much more poorly, and also are of course the earthquakes that are actually important to rapidly detect and alert from.

The authors do show the results of their model generalizing to larger finite-fault scenarios, modeled as a sequence of point sources, on which the model has not been trained on, which is impressive and does demonstrate the authors' model's "substantial potential to generalize effectively to finite-fault earthquake simulations". But since this is by far the most important quality of an EEW system, I believe that more emphasis should be put on this question, including training and testing the model on larger earthquakes in a more systematic manner.

3) Last but not least, the paper is in dire need of a real application. As it stands, their results show no indication that their approach works in real world scenarios, and to be the devil's advocate, their deep learning model may simply have memorized the simulated training data, especially since the testing data is so close to the training data (as I pointed out in 1). The authors mention that this is work in progress, but in my opinion the work is not finished without at least some initial results that show how their deep learning model performs on predicting ground motion for a real earthquake.

To summarize, the paper is very interesting, innovative, and quite convincing (with the reservation that the test set may be too 'easy' because too similar to the training data) but reads like somewhat unfinished work without at least some exploration of how it performs on real data.

I note that, unless I missed it, there are no code or data availability statements.

(Remarks on code availability)

Reviewer #2

(Remarks to the Author)

Kyu et al. present a Deep Learning algorithm (called WaveCastNet) trained to forecast the wavefield produced by synthetic earthquakes in the San Francisco Bay area using the first 15.6 seconds of seismic records. The approach is interesting and shows impressive performances for low magnitude synthetic events modelled as point sources and neglecting noise. However, the aforementioned conditions make it unsuitable for operational earthquake early warning applications (see detailed comments and questions below). The way I see it, this work is very preliminary and would need a lot more work (most of it being mentioned in the Perspective section) before making any claim regarding early warning applications. Therefore, I would not recommend this article to be published in Nature Communications.

Detailed comments and questions

(1) The authors mention that they “derive sensor locations from waveforms recorded over past decade”. Does it mean that the sensor selection is not based present station coverage? The considered stations should only be those presently transmitting in real time. Is it the case? If not, what would be the performance using only those stations.

(2) Noise should be added to the waveforms (both for training and testing). The most realistic strategy is to compile empirical noise, collected on the considered stations for the same time windows in all stations (in order to preserve the structure of network scale correlated noise). This is a redhibitory point in my mind. I expect that it will drastically deteriorate the performance of the algorithm, as it transforms the problem from a synthetic deterministic one (consisting in learning the imposed Green's functions) to a real-world underdetermined (and likely chaotic) one.

(3) Related to the previous comment, there is no detail on the considered set of earthquake rupture scenarios. The only mention is that “source parameters such as slip, slip rate, rupture initiation time, and local dip exhibit spatial variability and include stochastic fluctuations at minor scales.” To be realistic the source time functions should be drawn from realistic empirical distributions (which actually have fluctuations at major scales). I would suggest to use source time functions based on empirical observations (Meier et al., 2017). The same comment can be made for the slip distributions: slip distributions should be representative of empirical ruptures. Accounting for the exhaustiveness of possible earthquake scenarios will likely drastically deteriorate the performances.

(4) 0.5 Hz seems very low frequency. Peak ground velocity is expected to be significantly underestimated using such low-pass filtering (especially for low magnitude earthquakes). This may actually explain the apparent good performances for small events.

(5) For early warning applications, there is a set of additional issues to consider that are not discussed at all. The first phase of an early warning system is the detection. How is this phase intended to be performed with the proposed algorithm? Is the algorithm meant to be run continuously or triggered by a detection algorithm? How would the algorithm know time 0 (the time of the earthquake) necessary to feed the algorithm with the correct time windows (the one used in the training)? How would the algorithm deal with uneven latencies at the different stations (real-time data are transmitted in packets at different times with a few seconds intervals)? Considering all these technical issues, what kind of warning times would it provide? Would those warning times be useful? And how would they compare to existing approaches?

(6) Finally, there is no test on real-world data. The algorithm needs to be tested on real data.

All that considered, the algorithm seems too far to me from potential operational applications to consider publication in a high-impact journal. If the authors perform the extra-work I mentioned above and justify a gain in performance compared to existing systems, I would be happy to revise my judgement.

References

Meier, M. A., Ampuero, J. P., & Heaton, T. H. (2017). The hidden simplicity of subduction megathrust earthquakes. *Science*, 357(6357), 1277-1281.

(Remarks on code availability)

The link does not work.

Reviewer #3

(Remarks to the Author)

Review for- WaveCastNet: An AI-enabled Wavefield Forecasting Framework for Earthquake Early Warning

The authors here in this paper propose a novel AI-enabled framework, WaveCastNet to improve early warning by forecasting ground-motions from large earthquakes. WaveCastNet integrates a novel convolutional Long Expressive Memory (ConvLEM) model into a sequence-to-sequence (seq2seq) forecasting framework to model long-term dependencies and multi-scale patterns in both space and time. By sharing weights across spatial and temporal dimensions, WaveCastNet

requires fewer parameters compared to more resource-intensive models like transformers, thus reducing inference times. Moreover, WaveCastNet generalizes better than transformer-based models to various seismic scenarios, including rare and critical situations with higher magnitude earthquakes.

Minor Comments:

I would like to draw the authors' attention to some of the references that have been cited throughout the introductory text of this work. It seems like there are multiple areas in the manuscript where references are a tad amiss with the direction of the text.

1. Introduction Paragraph 2: "Inaccuracies in the parameter estimation, most commonly in over/under predictions in earthquake magnitudes, lead to false alert or missing warning opportunities [55, 42]." Although these papers are loosely related to the above statement, it does not really discuss over/under predictions in the way this paper does:

Avigyan Chatterjee, Nadine Igonin, Daniel T. Trugman; A Real-Time and Data-Driven Ground-Motion Prediction Framework for Earthquake Early Warning. *Bulletin of the Seismological Society of America* 2022;; 113 (2): 676–689. doi: <https://doi.org/10.1785/0120220180>

2. Introduction Paragraph 2: "The conventional use of empirical ground motion models precludes high fidelity representation of the complex source and path effects, and the site-specific variability of ground motion intensities [26, 8, 9, 14, 6]." I agree with the statement, but I am not sure how any of these citations are relevant to it. Most of the cited empirical ground motion models do not actually model their equations based on 'complex source and path effects'. In contrast, the representation of the source is very straightforward. I would ask the authors to look into papers like:

Wirth E. A. Vidale J. E. Frankel A. D. Pratt T. L. Marafi N. A. Thompson M., and Stephenson W. J. 2019. Source-dependent amplification of earthquake ground motions in deep sedimentary basins, *Geophys. Res. Lett.* 46, 6443–6450, doi: <https://doi.org/10.1029/2019GL082474>.

Sahakian V. J. Baltay A. Hanks T. C. Buehler J. Vernon F. L. Kilb D., and Abrahamson N. A. 2019. Ground motion residuals, path effects, and crustal properties: A Pilot study in southern California, *J. Geophys. Res.* 124, 5738–5753, doi: <https://doi.org/10.1029/2018JB016796>.

Parker G. A., and Baltay A. S. 2022. Empirical map-based nonergodic models of site response in the Greater Los Angeles area, *Bull. Seismol. Soc. Am.* 112, 1607–1629, doi: <https://doi.org/10.1785/0120210175>.

if they want talk about complex source and path effects.

3. Introduction Paragraph 3: "Artificial Intelligence (AI) provides a promising alternative approach for modeling ground motion propagation. That is because deep neural networks are well posed to model the nontrivial spatiotemporal properties of ground motions [19, 61, 11, 60, 20, 22, 57]. Moreover, AI methods have the advantage of being computational efficient during inference time, which is of great importance for early warning systems." Multiple citations in this statement are not related to AI in ground-motions at all.

61: Deep neural based method for phase picking.

57: Full waveform simulation method based on deep learning algorithms.

11: Related to phase picking.

I suggest citing these works instead:

O. M. Saad, I. Helmy, M. Mohammed, A. Savvaidis, A. Chatterjee and Y. Chen, "Deep Learning Peak Ground Acceleration Prediction Using Single-Station Waveforms," in *IEEE Transactions on Geoscience and Remote Sensing*, vol. 62, pp. 1-13, 2024, Art no. 5907213, doi: 10.1109/TGRS.2024.3367725.

Hsu, Ting-Yu, and Atteroni Pratomo. "Early peak ground acceleration prediction for on-site earthquake early warning using LSTM neural network." *Frontiers in Earth Science* 10 (2022): 911947.

4. Figure 1: The point source placements are hard to see and so are the stations in black. I think the scale of the figure can be adjusted a bit to make things more obvious (probably use a slightly bigger projection scale). Use a degree of transparency if needed to plot stations and event locations if needed.

The authors say that the red lines in the figure are known faults. Are they quaternary faults from the USGS fault database? Maybe good to mention. In Fig. 1b, the visco-elastic wave propagation at $T=21.79s$ is shown. I think it would nice to plot a colorbar of the wavefield velocity in the figure somewhere. Given it's the first figure, it might not be obvious to the reader what the color scale of the wavefield velocity is.

The colorbars for the rest of the figures have been disconcertingly placed. I would recommend placing them horizontally below similar panels.

5. Data and Code availability statement?

Major Comments:

1. The authors present their results and compare to it to existing seq2seq algorithms and, this present algorithm performs much better. However, most existing earthquake early warning systems still rely on traditional empirical ground motion models for the calculation of predicted ground motions. How does wavecastnet compare to empirical ground motion models for this region, both in terms of intensity values(PGA or PGV) and in terms of time?

2. Can the authors comment on how does local fault geometry affect the wavefield propagation and subsequent generation of ground motions? There has been some recent work led by researchers Victor Tsai, Greg Hirth and Daniel Trugman that talks about the influence of fault geometry.

<https://doi.org/10.1029/2024GL109418>

<https://www.nature.com/articles/s41586-024-07518-6>

Can you comment on how the presence of a complex fault geometry affect these simulations? SW4 takes into account topography effects during its simulations. Was that factored in?

(Remarks on code availability)

Data and Code availability statement missing.

Version 1:

Reviewer comments:

Reviewer #1

(Remarks to the Author)

The authors have addressed all my concerns, and made an impressive amount of additional work to prove that their model does generalize well to simulated finite fault ruptures, as well to a real world example, a Mw 4.4 earthquake.

I would like to see the limitations of the study explained in the rebuttal more clearly laid out in the introduction and conclusion of the main text, but I do recommend publication without further review.

(Remarks on code availability)

Reviewer #2

(Remarks to the Author)

The authors made a very significant effort to address the comments made by all 3 reviewers. I still think that, as now partially acknowledged by the authors, the proposed approach is very far from any potential operational earthquake early warning (EEW) application. I see it as a sophisticated deep learning model capable of learning the wavefield produced by a point source in the studied area. This is an interesting result by itself worthy of publication.

Overall, the manuscript is a good machine learning paper presenting an interesting application. As an EEW paper, it has significant flows, starting with the literature. Machine-learning-based early warning algorithms do exist (e.g., Li et al., 2018, Münchmeyer et al., 2021ab, Lin et al., 2021, 2023, Licciardi et al., 2022, Hourcade et al., 2025). One of them (Lara et al., 2023) is even implemented in an operational EEW system (Lara et al., 2025). The novelty proposed in this study is to model the entire wavefield and therefore directly the peak ground velocity rather than estimating the magnitude and location to indirectly assess the peak ground acceleration (side note: the authors should justify the choice of PGV rather than PGA, which is much more standard to assess damage).

For many technical reasons, I firmly believe that such an approach will never be of any operational use. Present EEW system strategies tend to minimize data transmission and computation as much as possible to both minimize latency and maximize resilience (data transmission in times of a catastrophic event is actually a big problem). The proposed approach will never fit in those strategies. The proposed approach is also subject to the same fundamental issues that classical EEW algorithms face: the information in the first seconds of the recorded seismograms being insufficient to characterize the full extent of the ongoing rupture (because it's not finished yet), EEW algorithms typically perform poorly for Mw ≥ 7 earthquakes. This approach does not fix the problem and extending the tests to Mw 5.5 events is not very convincing since earthquakes typically start producing significant damage for Mw ≥ 6 .

It is also worrying that the one real-data example the authors added fits much more poorly than all synthetic examples. This was to be expected (as I anticipated in my first review) but one should expect that for larger magnitude events (which are the ones we care about for EEW), this trend will be much accentuated because large magnitude earthquakes deviate a lot more from the synthetic sources considered in the training phase than the small ones (for which the point source approximation is much more realistic).

All that said, given the hard work that the authors put on to obtain the original results and to address all comments, and given that, as I said earlier, the deep learning model capturing the wavefield is a notable achievement worthy of publication by itself, I do not object to accept this paper provided that the authors put all claims about EEW into context (in terms of how it relates to existing approaches and in terms of limitations).

I will be happy to give a quick feedback on a revised manuscript rephrased accordingly.

References

- Hourcade, C., Juhel, K., and Bletery, Q. (2025). PEGSGraph: a Graph Neural Network for fast earthquake characterization based on Prompt ElastoGravity Signals. *Journal of Geophysical Research: Machine Learning and Computation*, 1, e2024JH000360.
- Lara, P., Bletery, Q., Ampuero, J. P., Inza, A., and Tavera, H. (2023). Earthquake Early Warning starting from 3 s of records on a single station with machine learning. *Journal of Geophysical Research: Solid Earth*, 128(11), e2023JB026575.
- Lara, P., Tavera, H., Bletery, Q., Ampuero, J. P., Inza, A., Portugal, D., ... and Meza, F. (2025). Implementation of the Peruvian Earthquake Early Warning System. *Bulletin of the Seismological Society of America*, 115(1), 191-209.
- Li, Z., Meier, M. A., Hauksson, E., Zhan, Z., and Andrews, J. (2018). Machine learning seismic wave discrimination: Application to earthquake early warning. *Geophysical Research Letters*, 45(10), 4773-4779.
- Licciardi, A., Bletery, Q., Rouet-Leduc, B., Ampuero, J. P., and Juhel, K. (2022). Instantaneous tracking of earthquake growth with elastogravity signals. *Nature*, 606(7913), 319-324.
- Lin, J. T., Melgar, D., Thomas, A. M., and Searcy, J. (2021). Early warning for great earthquakes from characterization of crustal deformation patterns with deep learning. *Journal of Geophysical Research: Solid Earth*, 126(10), e2021JB022703.
- Lin, J. T., Melgar, D., Sahakian, V. J., Thomas, A. M., and Searcy, J. (2023). Real-time fault tracking and ground motion prediction for large earthquakes with HR-GNSS and deep learning. *Journal of Geophysical Research: Solid Earth*, 128(12), e2023JB027255.
- Münchmeyer, J., Bindi, D., Leser, U., and Tilmann, F. (2021a). The transformer earthquake alerting model: A new versatile approach to earthquake early warning. *Geophysical Journal International*, 225(1), 646-656.
- Münchmeyer, J., Bindi, D., Leser, U., and Tilmann, F. (2021b). Earthquake magnitude and location estimation from real time seismic waveforms with a transformer network. *Geophysical Journal International*, 226(2), 1086-1104.

(Remarks on code availability)

Reviewer #3

(Remarks to the Author)

I am satisfied with the changes the authors have made. I agree with the authors that no manuscript/research is 100% complete and I appreciate the time and effort invested by the authors to address the questions and concerns raised. The only comment I would like to add here, is that when the authors are discussing future steps, it would be nice to add some more details as to how the present work can be adapted to real world scenarios. The authors have noted in their manuscript and their response file limitations with regards to mainly compute capabilities. A little more direction and discussion on the specifics might be useful.
I recommend accepting the manuscript for publication.

(Remarks on code availability)

Version 2:

Reviewer comments:

Reviewer #2

(Remarks to the Author)

I am satisfied with the last version of the manuscript that the authors provided and do not have further comments.

(Remarks on code availability)

Response to the comments of the reviewers on “WaveCastNet”

Dear reviewers,

We are very grateful for your careful reviews and insightful suggestions. These comments have provided us with valuable perspectives and have helped improve our method and manuscript considerably. We have developed a new methodology and added a number of new examples, figures, and discussion points to the manuscript. We have attempted to address each of the comments throughout the manuscript.

No single method is a silver bullet, especially when it comes to addressing early warning (EEW) challenges. Our hope is that the community can build on the WaveCastNet framework to develop more effective AI-enabled solutions to improve real-time predictions and improve resilience in the face of earthquakes. Although WaveCastNet is not yet ready to replace existing EEW systems, we believe it represents an important first step toward integrating machine learning with seismic forecasting. This paper focuses primarily on developing a novel deep learning architecture as a potential backbone for future EEW systems, rather than engineering a fully operational EEW solution. We recognize that expecting a single research paper to address all challenges in AI-based EEW systems is unrealistic. A key limitation, for example, is the availability of high-quality curated data for training AI models. We have invested significant resources in simulating realistic ground motion data and empirically evaluating the proposed model. Although we acknowledge that many additional scenarios must be considered, we believe that this will require years of further work. Similar traditional EEW systems have not been developed in a single research paper, and we view this as an important starting point for future research in this field.

To put this into perspective, considerable effort is being made in the field of AI for weather forecasting, where similar challenges are faced. Recent articles [5, 12, 19] published in high-impact journals such as Nature have focused on training AI models exclusively on simulated data, aiming to demonstrate the potential and advantages of machine learning in this domain, rather than trying to replace existing operational weather forecasting systems. These studies are framed as important first steps towards integrating AI into established forecasting frameworks, highlighting the need for further development and validation before practical implementation. Similarly, we view our work with WaveCastNet as a high-impact early stage contribution that

demonstrates the feasibility and promise of AI-based solutions for earthquake early warning, while recognizing that more research and validation is necessary.

We would like to stress that one of the key contributions of this work is the development of a novel deep learning architecture, the Convolutional Long Expressive Memory (ConvLEM) model, which is specifically designed to address the unique challenges of forecasting ground motions from large earthquakes. Rather than simply applying an existing model to a new problem, we have created a new architecture tailored to capture long-term dependencies and multi-scale patterns in seismic data. By incorporating convolutional layers with long expressive memory, our model effectively handles spatial and temporal correlations in the data, outperforming traditional transformer-based models. This development represents a significant step forward in seismic forecasting and sets the foundations for future AI-enabled approaches to EEW that are more efficient and better suited to the complexities of real-world data.

If reviewers feel that the current title may overstate the capabilities of our model, we are happy to modify it. A revised title could be “WaveCastNet: An AI-enabled Wavefield Forecasting Framework for Ground Motions from Large Earthquakes”, which reflects the scope of our work as a step towards AI-based earthquake forecasting.

Below is a brief summary of our response to the key concerns raised:

- **Real-world data evaluation:** We included new results using real seismic data, demonstrating WaveCastNet’s ability to generalize from synthetic simulations to real-world scenarios.
- **Demonstration of generalization beyond memorization:** We provided results that illustrate WaveCastNet’s ability to generalize across various seismic scenarios, including those not seen during training. These findings emphasize the model’s capacity to predict ground motions based on learned patterns rather than relying on the memorization of specific datasets.
- **Additional results for larger-magnitude finite-fault scenarios:** We include new analyzes that evaluate WaveCastNet performance in finite fault scenarios, demonstrating its ability to handle more realistic and complex earthquake rupture patterns, further validating its practical applicability.
- **Introduction of a new masked autoencoder approach:** We incorporated a novel masked autoencoder methodology into WaveCastNet, further enhancing its ability to handle sparse and noisy seismic data corresponding to the stations used in ShakeAlert. This addition demonstrates improved model performance in realistic scenarios and contributes to the growing body of work on efficient data encoding for seismic applications.

- **Model robustness:** Additional testing under various noise conditions and simulated uneven station latencies showed strong model performance, with limitations that arise when latencies exceeded certain thresholds.
- **Operational feasibility:** WaveCastNet operates continuously and does not require external triggers or precise time-zero information, distinguishing it from traditional EEW systems.
- **Comparison to existing EEW systems:** Although WaveCastNet is not yet a fully operational EEW system, it explores the potential of AI for earthquake forecasting. Direct comparisons to APPLE and PLUM were not possible due to their inaccessibility.
- **Frequency range and scalability:** We clarified that WaveCastNet can be extended to higher frequencies, although the computational resources and data required for demonstration remain a challenge.
- **Comparisons with empirical ground motions:** We perform comparisons with a ground motion model. We acknowledge that the current framework cannot directly validate PGVs due to the frequency range. Spectral acceleration is used instead.

We believe that the changes made, especially those related to real-world testing, noise resilience, and model robustness, significantly improve the manuscript. We are confident that the changes address the key concerns raised by the reviewers and improve the clarity and depth of our arguments.

Best Regards,

N. Benjamin Erichson, Dongwei Lyu, Rie Nakata, Pu Ren, Michael W. Mahoney, Arben Pitarka, and Nori Nakata

1 Referee 1

General author response: We sincerely thank the reviewer for their thoughtful and supportive assessment of our work. We are encouraged by your recognition that our results are convincing and that our specifically designed deep learning architecture demonstrates strong performance in forecasting ground motion.

We also fully agree with your observation that several important issues require further discussion and clarification. We are especially grateful for your clear summary of these concerns, which has helped us identify areas where the manuscript could be strengthened.

In response, we have carefully expanded the manuscript to address these issues in detail, providing additional analyses, discussions, and validations to ensure that the work is as comprehensive as possible.

=====

1. The paper does not elaborate at all about how the deep learning models are assessed in terms of training and testing. In particular, if testing and training rupture scenarios are assigned randomly, while all the sources are distributed along a dense line, all the model has to do is to interpolate between two nearby cases, and the model can simply ‘memorize’ the data. This concern could be alleviated by having test data coming from clearly distinct scenarios, for example the southern half of the fault used for training and the northern half used for testing.

Author response: We appreciate the reviewer bringing up this critical point regarding the assessment of our deep learning model and the potential for memorization. We have added a description of the train-test split in the Results section of the main text to improve clarity. We acknowledge that the nature of our training and testing setup could allow for interpolation between nearby scenarios if not carefully designed.

Memorization vs. Generalization in Deep Learning Models: It is important to discuss the issue of memorization in deep learning models in the context of generalization. The primary goal of any AI researcher is to develop a “useful” model capable of generalizing to new, unseen situations during inference. Memorization [4], on the other hand, refers to a model’s tendency to recall specific patterns or data points from its training set without necessarily understanding the underlying relationships or principles. Memorization is a natural aspect of any learning algorithm and is not inherently problematic, provided the model demonstrates an ability to generalize. This means that memorization is

acceptable as long as the model performs well on unseen data or scenarios that differ from the training distribution. Indeed, recent studies highlight that memorization often coexists with generalization [6], with the success of modern deep learning models resulting from their ability to balance the two.

In our case, WaveCastNet is designed to capture the spatiotemporal relationships governing ground motion propagation, rather than merely memorizing individual instances. Our extensive validations confirm that WaveCastNet goes beyond memorization, as evidenced by its ability to generalize to larger-magnitude finite-fault earthquakes, distinct spatial domains, and real-world events. In summary, while the model may exhibit some degree of memorization (a common behavior in machine learning) this is not a limitation in our study. Instead, WaveCastNet leverages its capacity to learn and generalize ground motion propagation, as demonstrated across diverse and challenging scenarios.

Measures Taken to Ensure Generalization Beyond Memorization:

- **Distinct Training and Testing Scenarios:** As described in Section 2.1, the training and testing rupture scenarios are derived from entirely distinct epicenters, exposing the model to a variety of source configurations. While the similarity in spatial locations of these epicenters may not entirely rule out interpolation, we have taken additional steps to address this concern, as outlined below.
- **Validation on Finite-Fault Events:** The model’s performance on finite-fault events (not seen during training), which involve complex kinematic finite-fault ruptures distinct from point-source earthquakes, highlights its ability to extrapolate beyond the specific patterns present in the training data. The integration of point-source responses over a fault plane introduces fundamentally new characteristics, further demonstrating that WaveCastNet learns underlying principles rather than memorizing individual cases.
- **Zero-Shot Generalization:** We evaluated WaveCastNet on a real-world event the Berkeley 2018 earthquake, which feature noise and complexities beyond those encountered during training. The model’s strong performance in these scenarios show its robustness, practical applicability, and capacity for generalization.
- **Distinct geographic regions:** Following the reviewer’s suggestion, we conducted additional experiments on earthquakes with epicenters located in regions clearly distinct from the simulation range. As shown in Figure

Figure 1: Peak Ground Velocity and Arrival Time maps for earthquakes with epicenters located entirely outside the simulation range (Event 1-Event 4).

1, the model maintains reasonable predictive performance for wavefields triggered by these out-of-range epicenters.

ACC	RFNE	RFNE	RFNE
		pgv	pga
0.75	0.64	0.30	0.29

Table 1: Average validation metrics for point-source earthquakes (Event 1–Event 4) outside of the simulation range, including their Peak Ground Velocity (PGV) and arrival time.

In summary, these validations, combined with experiments involving distinct geographic regions for training and testing, demonstrate that WaveCastNet is not simply interpolating between nearby scenarios but is capable of spatial extrapolation. This ability to generalize reinforces its potential as a predictive tool for earthquake early warning systems.

2. One of the most critical tasks for EEW is to correctly alert for larger earthquakes, that are rarer and for which EEW system always perform much more poorly, and also are of course the earthquakes that are actually important to rapidly detect and alert from.

The authors do show the results of their model generalizing to larger finite-fault scenarios, modeled as a sequence of point sources, on which the model has not been trained on, which is impressive and does demonstrate the authors’ model’s “substantial potential to generalize effectively to finite-fault earthquake simulations”. But since this is by far the most important quality of an EEW system, I believe that more emphasis should be put on this question, including training and testing the model on larger earthquakes in a more systematic manner.

Author response: We thank the reviewer for their insightful observation about the critical importance of accurately detecting and alerting for larger earthquakes in EEW systems. We appreciate the opportunity to further emphasize and expand on this point in our work.

Proof-of-concept: We would like to emphasize that this work represents one of the first efforts to apply AI-based models to the task of EEW. WaveCastNet is not intended as a direct replacement for existing operational EEW systems but rather as a proof-of-concept to demonstrate the potential of AI in this domain. By leveraging the strengths of deep learning, such as its ability to model complex spatiotemporal relationships and generalize across diverse scenarios, this study aims to showcase how AI could complement or enhance traditional approaches

in the future. Our focus has been on exploring the feasibility and robustness of this methodology, particularly its ability to generalize to larger earthquakes and complex fault dynamics, which are vital for improving the efficacy of EEW systems. We appreciate the reviewer’s feedback and see it as an opportunity to further refine and extend our approach toward this long-term goal.

Additional results for larger-magnitude finite-fault scenarios: In response to the reviewer’s suggestion, we have conducted additional testing to systematically evaluate the model’s performance on larger-magnitude finite-fault scenarios. Specifically, we performed new validation experiments using 10 finite-fault rupture scenarios for each magnitude level of earthquakes with moment magnitudes Mw 4.5–5.5. For these tests, we varied fault and hypocenter locations as well as the random-parameter seeds of the Graves-Pitarka kinematic rupture models. Importantly, these faults were positioned within the spatial domain covered by the point-source training data. As shown in Table 2, the accuracy (ACC) and relative Frobenius norm error (RFNE) metrics remained consistent with those reported in the main text. This highlights the robustness and reliability of the model’s generalization capabilities.

Mw	Fault Size (km × km)	T_{rup} (s)	ACC	RFNE	RFNE pgv	RFNE pga
4.5	1.8 × 1.8	3.5	0.94	0.39	0.10	0.17
5.0	3.4 × 3.0	3.7	0.93	0.41	0.12	0.18
5.5	8.0 × 4.0	6.0	0.94	0.56	0.33	0.23

Table 2: Average validation metrics for finite-fault simulations across all scenarios. The table presents results for different magnitudes (Mw 4.5–5.5), including accuracy (ACC), relative Frobenius norm error (RFNE) for the entire predicted wavefield sequences, and RFNE values for the predicted PGV and PGA maps

To further assess early warning performance, we have included additional metrics in Table 2, such as the relative error for the predicted peak ground velocity (PGV) and peak ground acceleration (PGA) maps. These results confirm the model’s capacity to generalize effectively to larger-magnitude finite-fault simulations, demonstrating its potential for practical applications in EEW scenarios.

Training on finite-fault data: We also acknowledge the reviewer’s point that training directly on finite-fault data could further enhance the model’s capabilities for larger earthquakes. However, generating realistic kinematic rup-

ture models for systematic training is a challenging task. It requires careful parameterization (e.g., rupture velocity, fault roughness, rise time) and validation of synthetic waveforms against observed data, which can be computationally intensive and non-trivial to scale effectively. Despite these challenges, we recognize the importance of this direction and identify it as a key area for future work.

Nevertheless, our results indicate that WaveCastNet’s ability to generalize to finite-fault cases, despite being trained exclusively on point-source scenarios, demonstrates its potential. The model does not rely on training for specific magnitudes or a priori knowledge of finite-fault versus point-source characteristics. This eliminates the need for magnitude estimates within the EEW framework and suggests that WaveCastNet could be directly applied to real-world scenarios without requiring magnitude-specific calibration.

In summary, while generating and systematically incorporating finite-fault data into the training process represents a valuable future step, the current results highlight that WaveCastNet already performs robustly for larger-magnitude earthquakes. We appreciate the reviewer’s suggestion and have expanded our discussion in the manuscript to emphasize this critical aspect of our work.

3. Last but not least, the paper is in dire need of a real application. As it stands, their results show no indication that their approach works in real world scenarios, and to be the devil’s advocate, their deep learning model may simply have memorized the simulated training data, especially since the testing data is so close to the training data (as I pointed out in 1). The authors mention that this is work in progress, but in my opinion the work is not finished without at least some initial results that show how their deep learning model performs on predicting ground motion for a real earthquake.

Author response: We thank the reviewer for highlighting the importance of demonstrating our model’s applicability to real-world scenarios. We agree that testing on real earthquakes is crucial for validating the practicality of our approach.

Results for real-world data: We conducted additional experiments to evaluate WaveCastNet on the 2018 Berkeley Mw 4.4 earthquake, recorded by 178 stations. This scenario presented an interesting challenge, as the input data was sparse and the model had been trained exclusively on synthetic point-source simulations. We explored two strategies for handling this real-world event:

- **One-Time Prediction Method:** This approach generates a full 110-second high-resolution sequence in one step. Sparse input data was initially processed by the sparse sampling model to predict the next 15.6 seconds of high-resolution wavefield, which was then input into the dense sampling model for the full sequence prediction.
- **Rolling Prediction Method:** This iterative method predicts the next 15.6-second segment using the current ground truth input and repeats the process for six steps to produce the complete 110-second sequence. While the prediction window is shorter, this method achieves better overall waveform consistency.

The results, summarized in Table 3, demonstrate that the model successfully reproduces the general waveforms, capturing both peak amplitudes and durations reasonably well, albeit with moderate accuracy for the wavefield.

Prediction - Setting	ACC	RFNE	RFNE
	wavefield	pgv	pga
One-time Prediction	0.38	0.28	0.26
Rolling Prediction	0.45	0.24	0.20

Table 3: Comparison of prediction accuracy and error metrics between the one-time prediction method and the rolling prediction method. Note that the wavefield target in this test is a 1D noisy signal, rather than the high-resolution 2D frames used in clean simulations. Therefore, the accuracy values presented here are not directly comparable to those from synthetic tests. Given the difficulty in achieving high accuracy for an entire waveform, we additionally evaluate relative error and accuracy of the peak velocity (pgv) across the station list and their arrival times (pga). These metrics provide a more practical assessment of the model’s performance as a part of an EEW framework.

Figure 2 shows the predicted high-resolution PGV amplitudes and arrival times, while Figure 3 visualizes the predicted waveforms at individual stations for both methods. These results showcase the model’s ability to handle real-world data and highlight its potential for deployment in EEW frameworks. Each method offers unique advantages: the one-time prediction method provides a longer prediction window, whereas the rolling prediction method ensures better consistency in waveform prediction.

Figure 2: High-resolution peak ground velocity and arrival time maps predicted for a real-world earthquake event.

Ground motion model comparisons To further validate the results, we compare our results with an ergodic ground motion model (GMM) using the ASK14 model developed as a part of the NGA-West2 project [3]. We use spectral acceleration instead of PGVs, since the maximum frequency of 0.5 Hz prevents us from evaluating PGVs properly. The PGVs calculated from this frequency range are much lower than the predictions from GMMs. In Figure 4, the spectral acceleration at 3 sec are shown for the GMM, predicted and groundtruth values. Since the WaveCastNet does not predict the first 15.6 waveforms, we compute spectral acceleration from the rest of the window. Hence at close to intermediate distances as shown in 3, the spectral acceleration from the prediction may not include an entire arrivals, which results in that WaveCastNet predictions are lower than GMM and groundtruths at distances smaller than 30 km. At large distances, we observe that the predictions are closer than

Figure 3: Predicted waveforms at two stations: San Jose (NP.1788, left) and Woodside (BK.JRSC, right). For NP.1788, both prediction approaches yield satisfactory results, with the rolling prediction showing notably higher accuracy. In contrast, the predictions for the velocity Z of NC.J026 show greater discrepancies.

Figure 4: Comparisons of the spectral acceleration at 3 sec with respect to distances between (blue) real-world data, (red) WaveCastNet predictions and (gray) empirical ground motion model predictions. Circles indicate the calculated spectral accelerations, and solid lines indicate their median values computed for bins of 10 km at 2 km intervals and moving-averaged between 3 nearby points.

the GMM predictions. This confirms that using WaveCastNet that are based on synthetic training data created from the carefully curated velocity model can

predict non-ergodic nature of ground motion intensities. This portion is added to Supplementary.

The challenge of domain shifts: Transitioning from synthetic simulations to real-world applications is challenging for deep learning models, as it involves discrepancies between synthetic and real waveforms. Such domain shifts are an active area of research in the broader AI community. For the San Francisco Bay Area (SFBA), synthetic waveforms generated using the USGS velocity model are reasonably accurate but not a perfect match with observed data [11, 16]. Addressing these discrepancies will be key to further improving the model’s real-world performance.

Fine-tuning: Future work will explore fine-tuning strategies to bridge the gap between synthetic and real data. For such strategies to be effective, the model must first be carefully pre-trained on a more diverse and representative dataset to enable better transfer to new and complex settings. While this effort is beyond the scope of the current work, it highlights the potential for developing a foundation model in the domain of earthquake early warning, with WaveCastNet serving as a promising backbone architecture.

To explore this potential, we conducted preliminary experiments to fine-tune our sparse model using six additional small-magnitude earthquakes. The fine-tuning process uses a mean squared error (MSE) loss function and a reduced learning rate, compared to the original learning rate. Initial results showed an improvement in the accuracy of wavefield predictions, with performance metrics increasing from 0.35 to 0.42.

These preliminary findings highlight the feasibility of fine-tuning as a means to enhance WaveCastNet’s performance under real-world conditions. Moving forward, however, we think it is required to develop a foundation model, with WaveCastNet serving as a promising backbone architecture. Such a foundation model could leverage pre-training on extensive synthetic and real-world data to enhance its adaptability and robustness across a wide range of scenarios.

In summary, while WaveCastNet’s current results on real-world earthquakes represent a proof of concept, they clearly demonstrate the feasibility of applying deep learning models like ours to practical EEW systems. We see the development of a foundation model in combination with fine-tuning with real-world data as an essential next step for improving model performance.

4. I note that, unless I missed it, there are no code or data availability statements.

Author response: The data will be made available after publication via NERSC data hosting, and the code is available at: <https://github.com/dwlyu/WaveCastNet>.

2 Referee 2

General author response: We sincerely thank the reviewer for their detailed feedback and critical assessment of our work. We appreciate the recognition of the novelty and impressive performance of WaveCastNet under the specific conditions evaluated in the study. We have carefully expanded the manuscript to address the important concerns raised, clarify the scope of this research, discuss its current limitations, and present solutions and directions for future development.

We would like to emphasize that our work represents one of the first AI-based approaches proposed for earthquake early warning (EEW) systems. Although we agree that this work is preliminary and has not yet achieved the level of maturity required for operational deployment, it serves as an important milestone in demonstrating the potential of deep learning for this critical application. WaveCastNet is not intended to replace existing EEW systems which have been developed and refined over many years, but rather to explore how AI techniques can complement and enhance these systems in the future.

Developing a fully operational EEW system is an ambitious goal that realistically requires a sustained and collaborative effort beyond the scope of a single research paper. However, we believe that this work establishes a strong foundation for the next generation of AI-enabled EEW systems by addressing key challenges such as wavefield forecasting and generalization to unseen earthquake scenarios.

In light of these contributions and the extensive revisions we have made, we believe that this work is a valuable and timely contribution to the field and warrants consideration for publication in Nature Communications.

=====

1. The authors mention that they “derive sensor locations from waveforms recorded over past decade”. Does it mean that the sensor selection is not based present station coverage? The considered stations should only be those presently transmitting in real time. Is it the case? If not, what would be the performance using only those stations.

Author response: We thank the reviewer for raising this important point. In our previous submission, we used a list of 564 stations based on historical data over the past decade. However, in response to your suggestion, we have now updated our station list to include only the 101 stations currently used for the Shake Alert system.

Masking for better handling of active ShakeAlert stations: To address the potential impact of station selection on model performance, we incorporated a Masked Autoencoder (MAE) approach during training on the original pool of 564 stations. The MAE framework allows us to reduce the number of active stations to 100 by employing a spatial-only masking strategy [7, 10]. In this strategy, a specific set of stations is masked for each training batch, but the mask remains consistent across time steps within each batch. This mimics real-world scenarios where the availability of station data remains stable over time, despite the dynamic nature of real-time sensor transmission.

We have updated all results in the manuscript to reflect the use of the ShakeAlert station list, and Table 4 now includes a comparison of evaluation metrics based on this new station set used during inference time.

Input - Setting	Number of Station	ACC	RFNE	RFNE	RFNE
				pgv	pga
Previous Station List	564	0.95	0.30	0.08	0.16
ShakeAlert Station List	101	0.93	0.36	0.11	0.16

Table 4: Performance metrics for WaveCastNet under sparse sampling scenarios, comparing the previous station list with the USGS real-time station list.

Moreover, by interpreting the masking strategy as a form of structural noise, uncertainty estimation as shown in Figure 5 for sparse sampling can be achieved with a single model by randomly generating multiple sets of masked stations during inference. This approach provides a more efficient alternative to ensemble methods that require multiple independently trained models in dense sampling scenarios.

Positive impact of MAE approach on real-world data scenarios: We would like to note that in our new experiments with real-world data, we found that when trained on a fixed set of 100 real-time stations, the standard WaveCastNet model achieved higher validation accuracy on simulated data. However, the MAE-based WaveCastNet model, which incorporates random station masking during training, demonstrated superior zero-shot generalization to real-world scenarios. This robustness in handling dynamic station availability suggests that the MAE-based model is more reliable.

- Noise should be added to the waveforms (both for training and testing). The most realistic strategy is to compile empirical noise, collected on the considered

Figure 5: Peak ground velocity and T_{PGV} map predictions with uncertainty quantification in sparse sampling scenarios.

stations for the same time windows in all stations (in order to preserve the structure of network scale correlated noise). This is a redhibitory point in my mind. I expect that it will drastically deteriorate the performance of the algorithm, as it transforms the problem from a synthetic deterministic one (consisting in learning the imposed Green's functions) to a real-world underdetermined (and likely chaotic) one.

Author response: We appreciate the valuable feedback from the reviewer regarding the addition of noise to the waveforms and agree that simulation of realistic noise conditions is an interesting scenario to evaluate the robustness of our model. In response, we conducted experiments to assess the performance of our model on noisy data, using the same task as in clean simulations: predicting a 100-second sequence from an initial 15-second input.

Additional results for simulating realistic noise scenarios: To simulate more realistic data scenarios, we consider additional experiments to study the impact of two types of noise into the input sequences:

- **Empirical Noise:** This noise was designed to preserve the network-scale correlations observed in real-world data. The empirical noise had the shape $C \times T(\text{Input Length}) \times N(\text{Number of Stations})$, where it was directly added to the original waveforms to simulate realistic noise patterns that occur in practice. It was generated by collecting empirical noise data from the stations in the same time windows as the seismic signals, ensuring that the noise characteristics matched real-world data.
- **Gaussian Noise:** To further stress test the model, we also applied Gaussian noise with shape $C \times T \times N$, where the noise values were sampled from a normal distribution $N(\mu = 0, \sigma = 0.32)$. The standard deviation σ of the Gaussian noise was chosen after several iterations to match the performance degradation observed with the empirical noise. This allowed us to evaluate the behavior of the model at varying levels ν of noise perturbation (that is, the noise level is a multiplicative factor $\sigma\nu$).

For both types of noise, we employed the ShakeAlert station list, which includes real-time stations, and used a sparse model trained without any noise injection to ensure the evaluations were realistic.

Our results, shown in Figure 6, demonstrate that the model maintains over 80% accuracy for both noise types at noise levels up to 2. This indicates that the model is resilient and can still perform well under moderate noise perturbations. We further illustrate this with a comparison of the model predictions at noise levels 1, 2, and 4, along with the corresponding PGV and PGA maps shown in Figures 7 and 8.

The robustness of our model can be attributed to the Seq2Seq-based architecture, where predictions are derived from a compressed context vector obtained from latent variables that are updated with each input step. This feature makes WaveCastNet highly efficient at denoising, showcasing its potential for real-world applications where noisy data is common.

We believe that these results demonstrate that our model is capable of handling the challenges posed by noisy environments.

3. Related to the previous comment, there is no detail on the considered set of earthquake rupture scenarios. The only mention is that “source parameters such as slip, slip rate, rupture initiation time, and local dip exhibit spatial variability and include stochastic fluctuations at minor scales.” To be realistic the source time functions should be drawn from realistic empirical distributions (which actually have fluctuations at major scales). I would suggest to use source

Figure 6: Evaluation metrics of WaveCastNet under varying levels of noise perturbation.

time functions based on empirical observations (Meier et al., 2017). The same comment can be made for the slip distributions: slip distributions should be representative of empirical ruptures. Accounting for the exhaustiveness of possible earthquake scenarios will likely drastically deteriorate the performances.

Author response: We appreciate the comments of the reviewer on the need for earthquake rupture scenarios to reflect realistic empirical distributions. To clarify, the kinematic rupture models used in our study were generated using the well-established physics-based Graves and Pitarka (GP) method [9, 8], which is widely recognized for its applicability in simulating both recorded and scenario earthquakes.

The GP method incorporates a realistic representation of source parameters, including slip, slip rate, rupture initiation time, and local dip, with spatial variability and stochastic fluctuations at minor scales. Furthermore, in our simulations, the slip time functions were improved following the methodology outlined by [18], providing a more nuanced representation of the rupture process.

We acknowledge the reviewer’s suggestion on the incorporation of source time functions and slip distributions based on empirical observations, such as those discussed in [15]. Although our current approach aligns with best practices in physics-based modeling, integrating empirical data to refine source-time func-

Figure 7: Waveforms at a single station (San Jose; NP.1788). The blue lines represent the ground truth, while the red lines of varying shades depict the input and predicted waveforms in both the time and frequency domains under different scales of noise perturbation for empirical noise (left) and Gaussian noise (right).

tions and slip distributions is indeed a logical next step. Empirical models can capture fluctuations at major scales and offer greater exhaustiveness in representing earthquake scenarios.

We also recognize that incorporating a more extensive range of possible earthquake scenarios, including empirically based models, could present additional challenges to model performance. However, we view this as an important area for future work to improve the robustness and generalizability of our approach in real-world settings. Exploring how AI models perform under these more comprehensive rupture scenarios will further validate their potential for earthquake early warning applications.

Finally, we would like to highlight that our use of the GP method, coupled with the high-performance computing environment [17, 13, 14], ensures a high level of realism in the rupture scenarios used in this study. However, we agree with the reviewer that the use of empirical distributions could further enhance the quality of the synthetic data set and provide a valuable extension of this

Figure 8: Peak Ground Velocity (PGV) and Arrival Time (PGA) maps for empirical noise (top) and white noise (bottom) across different noise levels.

work.

- 0.5 Hz seems very low frequency. Peak ground velocity is expected to be significantly underestimated using such low-pass filtering (especially for low magnitude earthquakes). This may actually explain the apparent good performances

for small events.

Author response: We appreciate the observation of the reviewer. To clarify, the primary objective of this study is to demonstrate the feasibility and potential of the proposed AI-based framework, WaveCastNet, for waveform predictions in earthquake early warning applications. The choice of frequency was guided by the computational constraints of simulating large-scale synthetic datasets, as well as the goal of validating the model in a controlled environment.

It is important to note that our method is not inherently constrained by this frequency range. The proposed Seq2Seq-based architecture and similar ML approaches have shown the ability to model high-frequency waveforms effectively, as evidenced by related work [21, 20].

Importantly, our new results for real-world data showcase the potential of WaveCastNet to generalize beyond such frequency limitations.

5. For early warning applications, there is a set of additional issues to consider that are not discussed at all. The first phase of an early warning system is the detection. How is this phase intended to be performed with the proposed algorithm? Is the algorithm meant to be run continuously or triggered by a detection algorithm? How would the algorithm know time 0 (the time of the earthquake) necessary to feed the algorithm with the correct time windows (the one used in the training)? How would the algorithm deal with uneven latencies at the different stations (real-time data are transmitted in packets at different times with a few seconds intervals)? Considering all these technical issues, what kind of warning times would it provide? Would those warning times be useful? And how would they compare to existing approaches?

Author response: We sincerely thank the reviewer for raising these important questions regarding the model’s applicability to early warning scenarios. Below, we address each point in turn:

- **Detection Phase:** The model’s architecture incorporates an implicit detection mechanism through its convolutional operations. Specifically, the model encodes relevant features from the input ground motion sequence into a latent space, which is then used to generate predictions. This encoding process serves as a substitute for a traditional detection phase. As noted in the abstract, our approach does not require explicit estimates of earthquake magnitudes or epicenters, and thus, does not include or rely on a separate detection phase. This feature differentiates WaveCastNet from traditional early warning systems.

- **Continuous Operation:** The algorithm is designed to operate continuously and iteratively, using a fixed-length input window of 15.6 seconds as described in the manuscript. When provided with an all-zero input sequence with minimal noise, the model generates predictions that are close to zero, effectively serving as a “quiet state.” Once significant ground motion begins, the model uses the early frames of input to predict future ground motion, providing an early indication of potential high-magnitude motion around the epicenter. Because the algorithm continuously processes data, it does not require knowledge of the earthquake’s exact start time or external triggers.
- **Warning Time and Practical Usefulness:** Figure 9 illustrates the relative prediction errors for wavefield and peak ground velocity (PGV) maps when the model operates continuously. The x-axis represents the elapsed time after the earthquake begins, during which the 15.6-second input sequence is updated with new data over time. Even with only 5 seconds of input data following the earthquake onset, the model produces reasonable predictions for waveforms and PGV. These predictions could offer meaningful lead times in real-world applications, particularly when paired with high-performance computing infrastructure.
- **Uneven Latencies Across Stations:** We acknowledge the challenge posed by uneven latencies in real-time data transmission. While the current version of WaveCastNet assumes synchronized input data, addressing latency variability is interesting. For instance, asynchronous data ingestion could be handled by developing a buffer or interpolation mechanism to align incoming packets with minimal delay.

Here, to address the issue of uneven latencies, we introduced random time shifts to all stations in the sparse sampling scenarios, as represented in Figure 10. The maximum time shift interval between stations was set to 1 second (equivalent to 4 discrete time steps). As shown in Table 5, the performance drop was minimal under these conditions. However, when the maximum time shift exceeded 2 seconds, we observed a notable decrease in accuracy, highlighting a current limitation of WaveCastNet. We acknowledge this as an important area for future work, as the model in its current form is not intended to serve as a fully-fledged EEW system. We appreciate the reviewer for emphasizing this critical direction for improvement.

- **Compatibility with Existing Systems and Frequency Range:** We have reached out to the developers of APPLE and PLUM, two systems that are most compatible with our method. Unfortunately, these systems are

Figure 9: Relative prediction errors for wavefield (left) and peak ground velocity (PGV) maps (right) when the model operates continuously. The errors change as the fixed-length input sequences are updated with new input data over time. The x-axis shows the elapsed time after the earthquake begins, representing the valid portion of the 15.6-second input sequence. Other components of the input are simulated as Gaussian noise.

Input - Setting	ACC	RFNE	RFNE	RFNE
			PGV	T_{PGV}
Without Time Shift	0.93	0.37	0.11	0.16
With Time Shift Between Stations	0.90	0.44	0.12	0.17

Table 5: Performance metrics comparison for WaveCastNet under time-shifted sampling scenarios.

not publicly available, preventing direct comparisons. However, we have included a comparison with an ergodic ground motion model (GMM) using the ASK14 model developed as part of the NGA-West2 project [3] in our newly added real-world example. For a more comprehensive discussion, we refer the reviewer to our response to Question 6.

We appreciate the reviewer’s valuable feedback and acknowledge that further refinements—such as handling data latencies and extending comparisons with existing early warning approaches—are critical steps for future work.

Figure 10: Peak ground velocity (PGV) map under time-shift conditions. The scatter points represent the shifted steps at each station.

Nonetheless, this study demonstrates the feasibility of an AI-driven approach for early warning, paving the way for more robust and adaptable systems.

6. Finally, there is no test on real-world data. The algorithm needs to be tested on real data.

Author response: We acknowledge that our previous manuscript did not include testing on real-world examples. In response to this feedback, we have incorporated real-world examples into our manuscript. We kindly direct the reviewer to the detailed discussion provided under item 3 in our response to Reviewer 1.

7. All that considered, the algorithm seems too far to me from potential operational applications to consider publication in a high-impact journal. If the authors perform the extra-work I mentioned above and justify a gain in performance

compared to existing systems, I would be happy to revise my judgement.

Author response: Thank you for your critical feedback on the operational applicability of our proposed framework. We want to clarify that this work is not intended to be directly deployed as an EEW system in its current form. We recognize that several crucial steps are required to bridge the gap between our approach and real-world operational needs, including extending the model to higher-frequency motions, improving its ability to handle large-magnitude events, and adapting it to meet real-time operational constraints.

It is important to note that this is a research paper aimed at developing and evaluating the potential of AI for predicting seismic waveforms, rather than an engineering effort to build a fully operational EEW system. Our key focus is the development of a novel deep learning model, rather than the engineering of a practical EEW system, which would require addressing additional complexities, such as robust real-time implementation, system integration, and comprehensive testing in diverse seismic conditions.

That said, we believe our work represents an important step forward in the field. The ability of our framework to predict waveforms, rather than solely ground motion intensity measures, is a novel and impactful contribution. The predicted waveforms provide a richer and more detailed dataset, enabling faster and more accurate assessments of critical infrastructure needs. This capability has significant potential to improve decision-making processes during destructive earthquakes.

8. Remarks on code availability: The link does not work.

Author response: The data will be made available after publication via NERSC data hosting, and the code is available at: <https://github.com/dwlyu/WaveCastNet>.

3 Referee 3

General author response: We sincerely thank the reviewer for their thoughtful evaluation of our work and for recognizing the novelty of our AI-enabled framework, WaveCastNet, for forecasting ground motions from large earthquakes. We appreciate the opportunity to address your comments and provide clarifications and additional information about our study.

We are encouraged by your recognition of WaveCastNet’s ability to generalize effectively across various seismic scenarios, including rare and critical high-magnitude events.

=====

1. I would like to draw the authors’ attention to some of the references that have been cited throughout the introductory text of this work. It seems like there are multiple areas in the manuscript where references are a tad amiss with the direction of the text.

Introduction Paragraph 2: “Inaccuracies in the parameter estimation, most commonly in over/under predictions in earthquake magnitudes, lead to false alert or missing warning opportunities [55, 42].” Although these papers are loosely related to the above statement, it does not really discuss over/under predictions in the way this paper does:

Avigyan Chatterjee, Nadine Igonin, Daniel T. Trugman; A Real-Time and Data-Driven Ground-Motion Prediction Framework for Earthquake Early Warning. Bulletin of the Seismological Society of America 2022;; 113 (2): 676–689. doi: <https://doi.org/10.1785/0120220180>

Author response: Thank you for the suggestion. Added to the reference.

2. Introduction Paragraph 2: “The conventional use of empirical ground motion models precludes high fidelity representation of the complex source and path effects, and the site-specific variability of ground motion intensities [26, 8, 9, 14, 6].” I agree with the statement, but I am not sure how any of these citations are relevant to it. Most of the cited empirical ground motion models do not actually model their equations based on ‘complex source and path effects’. In contrast, the representation of the source is very straightforward. I would ask the authors to look into papers like:

Wirth E. A. Vidale J. E. Frankel A. D. Pratt T. L. Marafi N. A. Thompson M., and Stephenson W. J. 2019. Source-dependent amplification of earthquake ground motions in deep sedimentary basins, Geophys. Res. Lett. 46,

6443–6450, doi: <https://doi.org/10.1029/2019GL082474>.

Sahakian V. J. Baltay A. Hanks T. C. Buehler J. Vernon F. L. Kilb D., and Abrahamson N. A. 2019. Ground motion residuals, path effects, and crustal properties: A Pilot study in southern California, *J. Geophys. Res.* 124, 5738–5753, doi: <https://doi.org/10.1029/2018JB016796>.

Parker G. A., and Baltay A. S. 2022. Empirical map-based nonergodic models of site response in the Greater Los Angeles area, *Bull. Seismol. Soc. Am.* 112, 1607–1629, doi: <https://doi.org/10.1785/0120210175>.

if they want talk about complex source and path effects.

Author response: We agree with the reviewer and acknowledge that we cited articles on ergodic ground motion models, which are the most commonly used in EEW systems. To address this, we have added references to highlight potential directions for incorporating non-ergodic ground motion models into EEW systems.

3. Introduction Paragraph 3: “Artificial Intelligence (AI) provides a promising alternative approach for modeling ground motion propagation. That is because deep neural networks are well posed to model the nontrivial spatiotemporal properties of ground motions [19, 61, 11, 60, 20, 22, 57]. Moreover, AI methods have the advantage of being computational efficient during inference time, which is of great importance for early warning systems.” Multiple citations in this statement are not related to AI in ground-motions at all.

61: Deep neural based method for phase picking. 57: Full waveform simulation method based on deep learning algorithms. 11: Related to phase picking.

I suggest citing these works instead:

O. M. Saad, I. Helmy, M. Mohammed, A. Savvaidis, A. Chatterjee and Y. Chen, "Deep Learning Peak Ground Acceleration Prediction Using Single-Station Waveforms," in *IEEE Transactions on Geoscience and Remote Sensing*, vol. 62, pp. 1-13, 2024, Art no. 5907213, doi: 10.1109/TGRS.2024.3367725.

Hsu, Ting-Yu, and Atteroni Pratomo. "Early peak ground acceleration prediction for on-site earthquake early warning using LSTM neural network." *Frontiers in Earth Science* 10 (2022): 911947.

Author response: Thank you for introducing these articles. These original citations are referred to as means in broader seismological contexts. We omitted phase-picking citations and have expanded and incorporated the suggested

references into the manuscript, along with several additional ML-based works, in the introduction to provide a more comprehensive context.

4. Figure 1: The point source placements are hard to see and so are the stations in black. I think the scale of the figure can be adjusted a bit to make things more obvious (probably use a slightly bigger projection scale). Use a degree of transparency if needed to plot stations and event locations if needed. The authors say that the red lines in the figure are known faults. Are they quaternary faults from the USGS fault database? Maybe good to mention. In Fig. 1b, the visco-elastic wave propagation at $T=21.79s$ is shown. I think it would nice to plot a colorbar of the wavefield velocity in the figure somewhere. Given it's the first figure, it might not be obvious to the reader what the color scale of the wavefield velocity is. The colorbars for the rest of the figures have been disconcertingly placed. I would recommend placing them horizontally below similar panels.

Author response: Figures 1, 2, 4 and 6 are changed accordingly to include transparency in Figures 1a,b, colorbar in Figure 1b, change the orientation of the colorbars in Figures 2,4,6. In Figure 1 caption, a reference to the USGS fault database is added, and we mention that the wavefield is in velocity.

5. Data and Code availability statement?

Author response: We will include a Data and Code availability statement in the final version. The data will be made available after publication via NERSC data hosting, and the code is available at: <https://github.com/dwlyu/WaveCastNet>.

6. The authors present their results and compare to it to existing seq2seq algorithms and, this present algorithm performs much better. However, most existing earthquake early warning systems still rely on traditional empirical ground motion models for the calculation of predicted ground motions. How does wavecastnet compare to empirical ground motion models for this region, both in terms of intensity values(PGA or PGV) and in terms of time?

Author response: Since our WaveCastNet predictions are very close to the physics-based simulation results (used as a groundtruth), the comparisons of the predictions of WaveCastNet prediction with the empirical ground motion models would actually mean the comparisons between the physics-based simulations and the empirical ground motion models, and hence evaluating the USGS velocity model. Comparisons have been carried out by many authors (including co-authors R.Nakata and A.Pitarka) up to 10 Hz [22, 1, 2, 16]. They demonstrated that stochastic characteristics (e.g. decay with distances) of the ground

motion intensities follow those of ergodic NGA-West2 ground motion models. They also showed that empirical and physics simulations differ at individual sites due to complex wave propagation effects, which cannot be modeled by ergodic ground motion models. Therefore, to avoid duplication of studies, we did not perform the comparisons for the synthetic motions in this study.

Instead, we compared with the ground motion models with the real-data example added in this revision. Please refer to the discussion under item 3 in our response to reviewer 1.

7. Can the authors comment on how does local fault geometry affect the wavefield propagation and subsequent generation of ground motions? There has been some recent work led by researchers Victor Tsai, Greg Hirth and Daniel Trugman that talks about the influence of fault geometry. <https://doi.org/10.1029/2024GL109418>
<https://www.nature.com/articles/s41586-024-07518-6>

Can you comment on how the presence of a complex fault geometry affect these simulations? SW4 takes into account topography effects during its simulations. Was that factored in?

Author response: In the Graves-Pitarka rupture generator, the small scale fault geometry spatial variations are used to emulate local spatial variations of the focal mechanism during an extended rupture. They only introduce correlated small perturbations to the local strike angle in addition to those of the rake angle, thus creating a less deterministic radiation pattern at short wavelengths which is qualitatively consistent with observations made for small earthquakes.

We do not incorporate the topography in the simulation, since the frequency of the simulation is low (<0.5 Hz), and since incorporating the topography increases the computational cost.

References

- [1] Brad T. Aagaard, Thomas M. Brocher, David Dolenc, Douglas Dreger, Robert W. Graves, Stephen Harmsen, Stephen Hartzell, Shawn Larsen, and Mary Lou Zoback. Ground-Motion Modeling of the 1906 San Francisco Earthquake, Part I: Validation Using the 1989 Loma Prieta Earthquake. *Bulletin of the Seismological Society of America*, 98(2):989–1011, 2008.
- [2] Brad T. Aagaard, Robert W. Graves, Arthur Rodgers, Thomas M. Brocher, Robert W. Simpson, Douglas Dreger, N. Anders Petersson, Shawn C. Larsen, Shuo Ma, and Robert C. Jachens. Ground-Motion Modeling of Hayward Fault Scenario Earthquakes, Part II: Simulation of Long-Period and Broadband Ground Motions. *Bulletin of the Seismological Society of America*, 100(6):2945–2977, 2010.
- [3] Norman A. Abrahamson, Walter J. Silva, and Ronnie Kamai. Summary of the ASK14 Ground Motion Relation for Active Crustal Regions. *Earthquake Spectra*, 30(3):1025–1055, 2014.
- [4] Devansh Arpit, Stanisław Jastrzebski, Nicolas Ballas, David Krueger, Emmanuel Bengio, Maxinder S Kanwal, Tegan Maharaj, Asja Fischer, Aaron Courville, Yoshua Bengio, et al. A closer look at memorization in deep networks. In *International conference on machine learning*, pages 233–242. PMLR, 2017.
- [5] Kaifeng Bi, Lingxi Xie, Hengheng Zhang, Xin Chen, Xiaotao Gu, and Qi Tian. Accurate medium-range global weather forecasting with 3d neural networks. *Nature*, 619(7970):533–538, 2023.
- [6] Satrajit Chatterjee and Piotr Zielinski. On the generalization mystery in deep learning. *arXiv preprint arXiv:2203.10036*, 2022.
- [7] Christoph Feichtenhofer, Haoqi Fan, Yanghao Li, and Kaiming He. Masked Autoencoders As Spatiotemporal Learners. In *Advances in Neural Information Processing Systems*, 2022.
- [8] Robert Graves and Arben Pitarka. Kinematic Ground-Motion Simulations on Rough Faults Including Effects of 3D Stochastic Velocity Perturbations. *Bulletin of the Seismological Society of America*, 106(5):2136–2153, 2016.
- [9] Robert W. Graves and Arben Pitarka. Broadband Ground-Motion Simulation Using a Hybrid Approach. *Bulletin of the Seismological Society of America*, 100(5A):2095–2123, 2010.

- [10] Kaiming He, Xinlei Chen, Saining Xie, Yanghao Li, Piotr Dollár, and Ross Girshick. Masked Autoencoders Are Scalable Vision Learners. In *Proceedings of the IEEE/CVF Conference on Computer Vision and Pattern Recognition*, pages 16000–16009, 2022.
- [11] Evan Hirakawa and Brad Aagaard. Evaluation and Updates for the USGS San Francisco Bay Region 3D Seismic Velocity Model in the East and North Bay Portions. *Bulletin of the Seismological Society of America*, 2022.
- [12] Dmitrii Kochkov, Janni Yuval, Ian Langmore, Peter Norgaard, Jamie Smith, Griffin Mooers, Milan Klöwer, James Lottes, Stephan Rasp, Peter Düben, et al. Neural general circulation models for weather and climate. *Nature*, 632(8027):1060–1066, 2024.
- [13] David McCallen, Anders Petersson, Arthur Rodgers, Arben Pitarka, Mamun Miah, Floriana Petrone, Bjorn Sjogreen, Norman Abrahamson, and Houjun Tang. EQSIM—A Multidisciplinary Framework for Fault-to-Structure Earthquake Simulations on Exascale Computers Part I: Computational Models and Workflow. *Earthquake Spectra*, page 875529302097098, 2020.
- [14] David McCallen, Floriana Petrone, Mamun Miah, Arben Pitarka, Arthur Rodgers, and Norman Abrahamson. EQSIM—A Multidisciplinary Framework for Fault-to-Structure Earthquake Simulations on Exascale Computers, Part II: Regional Simulations of Building Response. *Earthquake Spectra*, page 875529302097098, 2020.
- [15] M.-A. Meier, J. P. Ampuero, and T. H. Heaton. The Hidden Simplicity of Subduction Megathrust Earthquakes. *Science*, 357(6357):1277–1281, 2017.
- [16] Camilo Pinilla-Ramos, Arben Pitarka, David McCallen M. EERI, and Rie Nakata. Performance evaluation of the USGS velocity model for the San Francisco Bay Area. *Earthquake Spectra*, 41(1):457–494, 2024.
- [17] A. Pitarka, R. Graves, K. Irikura, K. Miyakoshi, and A. Rodgers. Kinematic Rupture Modeling of Ground Motion from the M7 Kumamoto, Japan Earthquake. *Pure and Applied Geophysics*, 177(5):2199–2221, 2020.
- [18] Arben Pitarka, Robert Graves, Kojiro Irikura, Ken Miyakoshi, Changjiang Wu, Hiroshi Kawase, Arthur Rodgers, and David McCallen. Refinements to the Graves–Pitarka Kinematic Rupture Generator, Including a Dynamically Consistent Slip-Rate Function, Applied to the 2019 Mw 7.1 Ridgecrest Earthquake. *Bulletin of the Seismological Society of America*, 2021.

- [19] Ilan Price, Alvaro Sanchez-Gonzalez, Ferran Alet, Tom R Andersson, Andrew El-Kadi, Dominic Masters, Timo Ewalds, Jacklynn Stott, Shakir Mohamed, Peter Battaglia, et al. Probabilistic weather forecasting with machine learning. *Nature*, 637(8044):84–90, 2025.
- [20] Pu Ren, Rie Nakata, Maxime Lacour, Ilan Naiman, Nori Nakata, Jialin Song, Zhengfa Bi, Osman Asif Malik, Dmitriy Morozov, Omri Azencot, N Benjamin Erichson, and Michael W Mahoney. Learning Physics for Unveiling Hidden Earthquake Ground Motions via Conditional Generative Modeling. *arXiv*, 2024.
- [21] Pu Ren, Rie Nakata, Osman Asif Malik, Nori Nakata, Dmitriy Morozov, N Benjamin Erichson, and Michael W. Mahoney. Physics-constrained Convolutional Recurrent Learning for Modeling Seismic Wave Propagation. *SEG*, 2024.
- [22] Arthur J. Rodgers, Arben Pitarka, N. Anders Petersson, Björn Sjögreen, and David B. McCallen. Broadband (0–4 Hz) Ground Motions for a Magnitude 7.0 Hayward Fault Earthquake With Three-Dimensional Structure and Topography. *Geophysical Research Letters*, 45(2):739–747, 2018.

Response to the comments of the reviewers on
“WaveCastNet”

1 Referee 1

General author response: We thank the reviewer for the positive and encouraging feedback, and for recognizing the additional work we conducted to demonstrate generalization to both simulated finite-fault events and real-world data. In response to the reviewer's final suggestion, we have now included a short discussion of the limitations of our approach at the end of the introduction. We appreciate the recommendation for publication and the constructive input throughout the review process.

=====

2 Referee 2

General author response: We would like to sincerely thank the reviewer for their additional thoughtful feedback and for acknowledging that the deep learning framework we propose, despite its current limitations, provides a significant technical contribution worthy of publication. We particularly appreciate the reviewer’s recognition of the effort we have made to respond comprehensively to earlier comments and to revise the manuscript accordingly.

The EEW problem is motivating our proposed AI-based approach, but we fully agree that this study should be understood as a proof-of-concept, not an operational EEW system. In line with the reviewer’s guidance, we have removed nearly all general references to EEW from the manuscript and integrated the suggested references on deep learning for EEW to better contextualize our work. This said, if the reviewer thinks that it is beneficial, we are also happy to change the title: “WaveCastNet: An AI-enabled Framework for Rapid Earthquake Wavefield Forecasting”.

We acknowledge the reviewer’s skepticism about the operational applicability of our approach to EEW, and feel that this discussion is important and fruitful. This said, we would like to respectfully offer a different perspective. We believe that the modeling paradigm introduced in this paper, namely, sequence-to-sequence forecasting of full wavefields using AI, holds long-term potential for integration into EEW systems, especially as computational infrastructure, data pipelines, and modeling strategies continue to evolve.

If we understand the reviewer correctly, then there is agreement that having access to the full wavefield is beneficial in principal. However, the key criticism of the reviewer is that it is too computational demanding and challenging to obtain the wavefield in an operational setting in real-time. We would like to challenge this concern, and support why we believe the system that we propose can be made operational.

- **Data Transmission:** We do not anticipate this being a fundamental limitation. Our model is designed to work effectively with sparse sensor inputs, and modern fiber-optic and low-latency communication infrastructures are increasingly capable of supporting the data demands of distributed sensor networks, particularly for small payloads such sparse measurements.
- **Computation and Latency:** Our current implementation achieves low-latency inference (0.56 seconds for 100 seconds of wavefield data) using an NVIDIA A100 GPU from 2020. With next-generation hardware (e.g., Nvidia’s Blackwell architecture) and further model optimization, we expect inference times to decrease significantly. Thus, we do not view computational cost as a

long-term bottleneck. Note that we have not used any of the state-of-the-art tricks for optimizing inference time, nor have we taken advantage of mixed-precision computation.

- **Generalization to Larger Magnitudes:** While our training set consists of magnitude M4 events, the model generalizes robustly up to M6 in synthetic tests. We believe that this is a promising result, particularly considering the limited diversity of the training data. Improved performance at higher magnitudes will likely come from broader training datasets that include more realistic finite fault simulations, which we highlight as a future direction. (Note that the bottlenecks are the cost and time to simulate these data. Thus, we currently only evaluated our model on these limited data, rather than using the data for training.)
- **Real-World Data:** We agree that performance on real data remains a key challenge. That said, we find the zero-shot generalization to a real event in the Berkeley dataset encouraging, especially considering that no real earthquakes were used during training. With additional work, such as domain adaptation or fine-tuning on real seismic data, we anticipate substantial improvements.
- **First-Seconds Information Limitation:** We agree that all real-time approaches to EEW are limited by the information content available early in a rupture. However, we are optimistic that AI-based methods can better leverage subtle and higher-dimensional patterns in the data than classical algorithms. Although this advantage has yet to be definitively demonstrated for large-magnitude events, it remains a plausible hypothesis supported by parallel advances in fields like weather forecasting and protein folding.
- **Choice of PGV over PGA:** We use PGV rather than PGA to enable direct comparison between the ground truth and forecasted data, both of which are velocity time series. Good agreement across a broad frequency range suggests that PGA would also show a similar level of agreement as PGV.

We view this work as a first step toward full wavefield forecasting using deep learning for ground-motion forecasting modules in EEW. As with any new modeling direction, intermediate results are necessary to build momentum, gather community feedback, and attract the resources needed for more ambitious development. Although we agree that our approach is not ready for operational deployment, we respectfully assert that dismissing its long-term potential may be premature. We believe that our work provides a strong foundation for future exploration and encourages broader thinking about how AI for full wavefield forecasting can support EEW. From a ML point of view, we don't see critical technical road blocks that would hinder its success.

3 Referee 3

General author response: We thank the reviewer for the thoughtful feedback and their supportive comments throughout the review process. We agree that further work is needed to bring this framework closer to real-world applications, and we appreciate the suggestion to elaborate on this in the manuscript.

In response, we have expanded the discussion of future directions to include more concrete steps for adapting our model to real-world scenarios. Specifically, we now highlight the importance of (1) expanding the training dataset to include more diverse and higher-magnitude events beyond M4, particularly finite-fault simulations; (2) fine-tuning on real seismic data to improve domain adaptation; and (3) scaling the model capacity to better capture the broader range of physical phenomena present in real events. These steps, combined with continued progress in computing infrastructure and simulation capabilities, will support the development of robust, deployable systems based on the proposed approach.

We thank the reviewer again for their recommendation for publication.

=====